Methods

# Endo-bind-n-seq: identifying RNA motifs of RNA binding proteins isolated from endogenous sources

Tiana Nicole Hanelt[1],*, Nora Treiber[1],*, Thomas Treiber[1],*, Gerhard Lehmann[1], Norbert Eichner[1], Tamara Rothmeier[2], Georg Schmid[2], Robert Reichelt[2], Federico Zambelli[3] , Giulio Pavesi[3], Dina Grohmann[2], Gunter Meister[1,4]

**RNA binding proteins (RBPs) are crucial regulators of gene expression and critically depend on the specific recognition of their target RNAs. Accordingly, a selection of methods to analyze RBP specificities has been developed, including protein-RNA cross-linking and sequencing (CLIP) and in vitro selection methods such as SELEX, RNA compete or RNA bind-n-seq. However, limitations like the availability for purified recombinant proteins and custom microarray platforms (RNAcompete) or extensive sequencing depth and sophisticated bioinformatic data processing (CLIP) may limit a broader implementation of these methods. Here, we present an RNA bind-n-seq method that uses short random RNA pools and enables multiple rounds of selection. This results in strong motif enrichment with low positional variance thus reducing sequencing depth requirements. Furthermore, we have coupled our protocol to immunoprecipitation of tagged or endogenous RBPs from cultured cells or tissue samples, eliminating the need for recombinant proteins. Our method also allows for the identification of indirect RNA motifs of proteins that are integral parts of multiprotein RNPs and result in physically more relevant RNA motifs.**

## Introduction

RNA molecules are central players in gene expression and regulation and form functionally diverse complexes (RNPs) with a variety of RNA-binding proteins (RBPs). Such RNPs range from rather dynamic mRNPs to large molecular machines like the spliceosome or ribosomes. To date, over 1,500 RBPs have been identified in human cells (Gerstberger et al, 2014; Hentze et al, 2018). Due to their fundamental roles in gene regulation, many RBPs have been associated with diseases and are increasingly targeted for therapy (Wu, 2020; Gebauer et al, 2021; He et al, 2023). To recognize and contact their specific RNA partners, RBPs possess diverse RNA binding domains (RBDs). Depending on their RNA target structures, RBDs can either bind dsRNA mainly by interacting with the major or minor grove (Masliah et al, 2013) or with single-stranded RNA by establishing specific contacts to mainly unfolded regions of the RNA. However, binding can also be more complex and RBDs may contact short, structured elements either by contacting the sugar-phosphate backbone or the bases (Corley et al, 2020). RBDs which target single-stranded RNAs are very common and are found in a large number of proteins. Examples of well-studied RBDs are the RNA recognition motif (RRM) (Daubner et al, 2013), the C3H zinc finger (Fasken & Corbett, 2005) or the hnRNP K-homology (KH) domain (Nicastro et al, 2015). Strategies to globally identify protein-RNA interactions such as mRNP capture approaches suggest that many so far unrecognized RBDs may exist (Baltz et al, 2012; Castello et al, 2012; Kwon et al, 2013). One example is the WD40-like NHL domain found in the TRIM-NHL protein family (Connacher & Goldstrohm, 2021). Several NHL domains bind RNAs highly specifically. Interestingly, target sites can either be structured (Kumari et al, 2018) or linear sequence motifs (Loedige et al, 2014, 2015) highlighting the broad binding spectrum of these WD40-like domains. RNA motifs of RBDs are often only 4–6 nucleotides in length. Since such short motifs are statistically very frequent in the human transcriptome, specificity of RBPs is often increased by a modular combination of either the same or different RBDs (Lunde et al, 2007; Hennig & Sattler, 2015). Such combinations can lead to highly complex and thus very specific target sites.

To understand the biological roles of RBPs, a detailed knowledge of their cognate RNA motifs is important. In RNA populations enriched in RNA immunoprecipitation experiments (RIP), sequence motifs can be extracted using algorithms such as MEME (Bailey et al, 2015). Depending on the affinity of the RBP to the target site, degeneration of the motif or usage of the correct cell type, retrieving RNA motifs from RIP data can be challenging. A further development of such experiments are combinations of RIP with UV-crosslinking steps, methods commonly referred to as Crosslinking Immunoprecipitations (CLIP) (Ule et al, 2003; Chi et al, 2009; Hafner et al, 2010). Here, RBPs are covalently crosslinked to their RNA target allowing for much more stringent washing steps. A potential

---

[1]Regensburg Center for Biochemistry (RCB), Laboratory for RNA Biology, University of Regensburg, Regensburg, Germany   [2]Regensburg Center for Biochemistry (RCB), Institute of Microbiology & Archaea Centre, Single-Molecule Biochemistry Lab, University of Regensburg, Regensburg, Germany   [3]Dipartimento di Bioscienze, Università di Milano, Milan, Italy   [4]Cluster for Nucleic Acid Therapeutics Munich (CNATM), Munich, Germany

Correspondence: gunter.meister@ur.de
Georg Schmid's present address is Microbify GmbH, Straubing, Germany
*Tiana Nicole Hanelt, Nora Treiber, and Thomas Treiber contributed equally to this work

---

target sequence is often stronger enriched and RNA motif determination is more accurate. Although highly enriched RNA motifs can be identified from CLIP approaches, such experiments are challenging in terms of experimentation and data analysis.

As an alternative and complementary approach, a panel of in vitro methods is available. In Systematic Evolution of Ligands by Exponential Enrichment (SELEX) experiments (Kohlberger & Gadermaier, 2022), recombinant RBPs are used to select a binding sequence from a pool of input sequences (library). Bound RNA fragments are eluted and analyzed for enriched motifs. These SELEX experiments are conducted in several rounds of selection and amplification to increase the enriched target sequence. This SELEX method usually employs random sequence inserts flanked by primer binding sites for the amplification cycles and is widely used for identification of aptamers – folded RNA molecules that can specifically recognize different types of interaction partners (Groher & Suess, 2016). But SELEX strategies using next generation sequencing readouts have also been reported (Cole & Luptak, 2019). The SELEX principle has been further adapted to RBPs in methods referred to as RNAcompete and RNA bind-n-seq to enrich for RNA motifs from a complex pool of input sequences. In RNAcompete, immobilized recombinant RBPs are incubated with an in vitro transcribed RNA library and after one single selection cycle, bound RNAs are eluted and hybridized to an array of the input DNA templates (Ray et al, 2009; Ray et al, 2013). RNA bind-n-seq resembles an adaptation of this protocol that uses synthetic RNA libraries and RNA sequencing after the selection round (Lambert et al, 2014).

The described in vitro approaches can be powerful but require recombinant RBPs, which might not always be available. Furthermore, one selection round appears to be sufficient, but the results might be statistically less significant, which could be problematic when low affinity RNA binders are investigated. In addition, in vitro methods using recombinant proteins might not always fully recapitulate the situation within a living cell. For example, RBPs might be in complex with other proteins, which might change their target binding affinity. Or, RBPs might be post-translationally modified, which could also affect their binding activity and finally, RNA target interactions might be dynamic and only biologically relevant during a short developmental window or in distinct tissues. To overcome these limitations, we improved the standard RNA bind-n-seq protocol. First, we introduced a second selection step to increase statistical power. Second, we demonstrate that the RNA motif selection can be performed on immunoprecipitated RBPs and protein complexes from various sources including mouse liver or brain tissues, allowing to preserve more natural RBP conditions. Finally, we determine the minimal protein amount that is needed to retrieve a reliable RNA motif. We refer to this improved method as RNA endo-bind-n-seq, which allows for the identification of RNA motifs from tissues or different cell stages.

# Results

## Two selection cycles increase the significance of enriched RNA motifs

To improve existing RNA bind-n-seq protocols, we first generated a random RNA library. Most RBDs recognize short linear sequence motifs of 4–6 nt (Hennig & Sattler, 2015) and we therefore reasoned that a random RNA pool of 8 nt should be sufficient to cover the recognition sites of most domains whereas reducing problems arising from RNA folding or multiple overlapping motifs. Thus, we generated an input pool containing eight randomized nucleotide positions followed by the four invariant bases GUUU to later allow for a second library generation. We validated the base distribution in the input pool by RNA-seq and optimized the synthesis conditions to reach a homogenous base distribution at all positions, which was unfortunately not fully achieved towards the 3′ end of the 8-mer library (Fig S1A), whereas the nucleotides of a 14-mer library were almost equally distributed (Fig S1B). The 14 nt library was generated for the analysis of RBP complexes or more extended RNA motifs.

To allow for two selection steps, we introduced several modifications to the existing bind-n-seq protocol (Fig 1A). The selected RNA is ligated to a 3′ DNA adapter, which is pre-adenylated by the 5′ nucleotide extension AppAAAC. Next, a 5′ RNA adapter is added, which contains a T7 RNA polymerase promoter at its 3′ end (Fig 1A). The ligation product is reverse transcribed, PCR amplified, and size selected on a denaturing acrylamide gel to enrich for insert-containing fragments (Fig S2A). The resulting first round-selected library is subjected to sequencing.

To enable a second round of selection, the library obtained during the first selection round is amplified in a scale-up PCR reaction and the PCR product is cleaved by the restriction enzyme MssI. This enzyme recognizes the sequence GTTTAAAC, which is generated from the invariant GUUU 3′ end of our starting RNA library and the 5′ extension of the 3′ adapter (Fig 1A). By catalyzing a blunt-end cleavage reaction between the two half sites, this step restores the 3′ end of the original input pool. The cleavage reaction is directly diluted with a T7 polymerase transcription mix to transcribe the selected inserts into a new RNA pool (Fig S2B), which matches the design of the input in the first binding reaction with the exception of a 5′GGG triplet (Fig 1A). This is encoded in the T7 promoter sequence and serves as optimal transcription start site, thus preventing a 5′ bias in the transcribed inserts. The resulting RNA pool can now be used for a second round of selection (Fig 1A).

We tested our method with two well-characterized, GST-tagged and purified recombinant RBPs: A N-terminal ZC3H10 fragment containing the three zinc finger domains and a coiled-coil region, and hnRNPA1 (Fig 1B). Using a modified version of the program Weeder2 (Zambelli et al, 2014), we strongly enriched the known RNA motifs of hnRNPA1 and ZC3H10 from the obtained sequence reads after two rounds of selection. HnRNPA1 enriches the motif UAGGGA (Fig 1B). This is in perfect agreement with the motifs found by SELEX (Burd & Dreyfuss, 1994) and RNA compete (Ray et al, 2013). ZC3H10 selection yields an AGUGCAG motif, which corresponds well to but is longer than the motif identified by RNA compete (GCAGCG). The additional G of the RNA compete motif corresponds to the invariant 3′ G in our input RNA and is removed from the reads before data analysis (see Fig S3 for the top-scoring Weeder output sequences).

We next tested a different tag and different beads material to further control for potential background binding effects (Fig 1C). We obtained identical motifs for the GST and hexa-histidine (His) fusions of the hnRNPA1 RRMs indicating robust and bead-independent motif

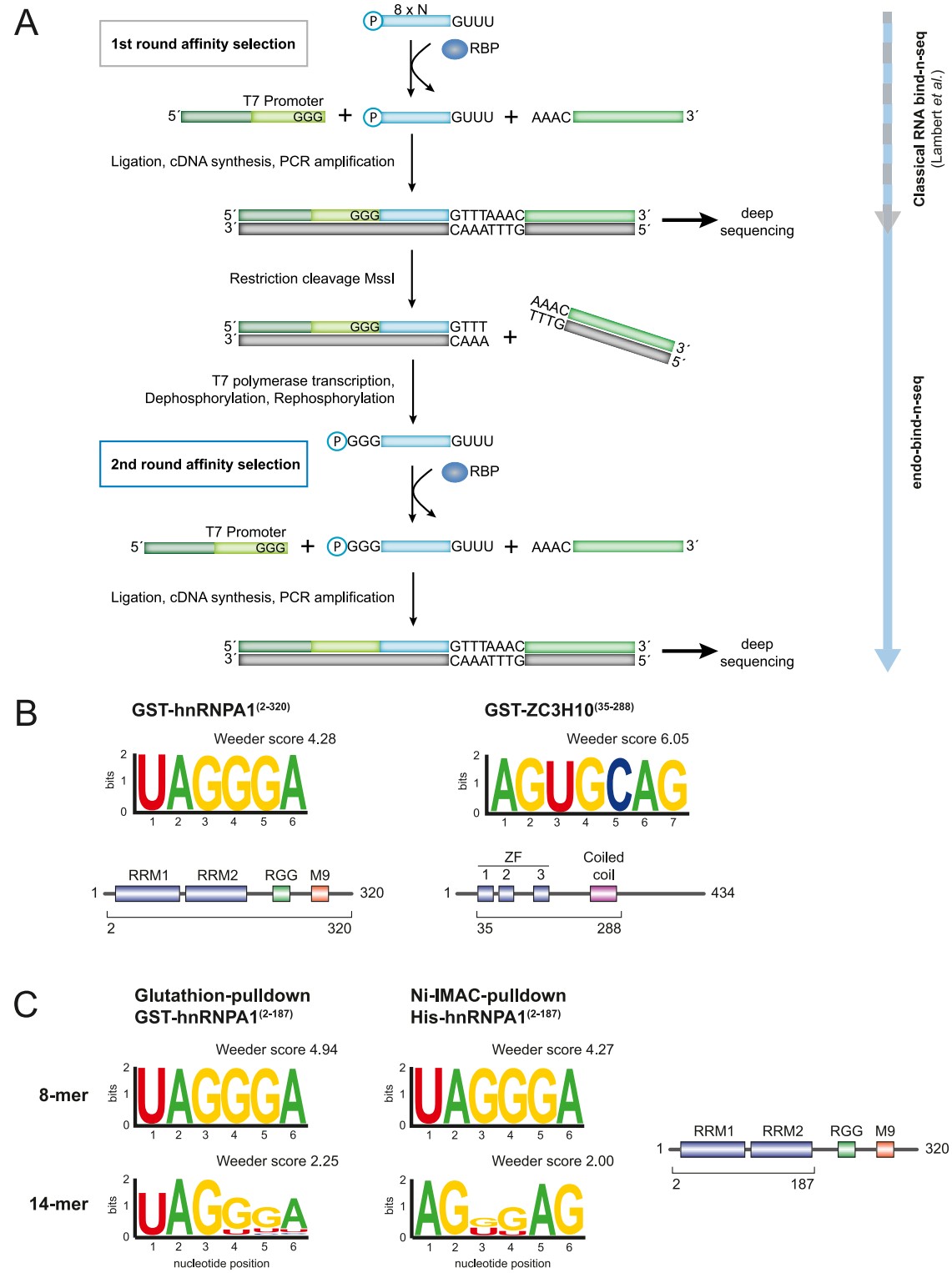

**Figure 1. Pipeline and validation of the endo-bind-n-seq protocol.**
**(A)** Schematic overview of the experimental method workflow. The investigated RBP is immobilized on a bead-based matrix and faced to a randomized RNA pool (blue, usually eight random nucleotides) for interaction. Protein associated target RNA motifs are isolated and cloned for deep sequencing (adapter in green). Resulting cDNA library is used as a template for run-off transcription. The 3' adapter is removed by MssI restriction digestion and a T7 RNA polymerase promoter site in the 5'-adapter (light green) is used for transcription of a new RNA pool. The new, specified pool, which harbors increased numbers of potential target RNA motifs of the RBP, is used in a second round of target RNA motif selection. **(B)** Sequence logo representation of enriched motifs obtained after two RNA target selection rounds, following the endo-bind-n-seq protocol. A randomized 8-mer RNA pool was incubated with indicated GST-fusion proteins. Enrichment scores from Weeder2 analysis are given (top). Purified

enrichment. In addition to the RNA library with eight random positions, we also tested the 14-mer library in the selection reactions with GST- or His-hnRNPA1 RBDs (Fig 1C). The enriched motifs in these datasets reached lower significance scores and are less clearly defined. This is most likely due to the presence of multiple overlapping binding sites that are present on one read. Sequences in such reads may be difficult to define and therefore scores drop. In addition, 14-mers might form more secondary structures, which could also affect data analysis. Taken together, the established two-round RNA bind-n-seq protocol is robust and identifies clear motifs from an 8-mer RNA library when recombinant RBDs are used.

### Identification of RNA motifs of so far uncharacterized RBPs

We next used our modified bind-n-seq protocol to identify so far unknown RNA binding activities. We have previously identified ZC3H7B as a specific interactor of the miR-7-1 precursor, where it binds to the terminal loop sequence (GAUAAC) (Treiber et al, 2017). Consistently, we find an enrichment of AUAG selected with the C-terminal domain of ZC3H7B, which contains five putative zinc fingers (ZC3H7B[415–956], Fig 2A). To validate our results with an independent approach, we performed filter binding assays. Recombinant ZC3H7B(415–956) shows a strong binding preference to an RNA fragment containing the identified AUAG sequence (AUAGUGUAGU) compared with a sequence with a mutant motif (AAAGUGAAGU) and an unrelated control (Fig 2B) corroborating our bind-n-seq results.

To demonstrate broad applicability of our approach, we investigated RNA binding activity of the archaeal RBP SmAP1 from *Pyrococcus furiosus* (Pfu). The recombinant protein was purified and immobilized using a Strep-tag, adding another tag and beads material to our study (Fig 2C). After two selection rounds, we enriched for the sequence motif UUGAGUU from the library. Again, using electro mobility shift assay experiments as independent approach, the identified motif bound efficiently and specifically, validating our bind-n-seq results (Fig 2D). Consistently, a U-rich motif has very recently been retrieved from RIP-seq experiments of Strep-tagged PfuSmAP1 (Reichelt et al, 2023).

Based on the Weeder scores, we finally examined whether a second round of selection is actually beneficial for motif selection. Indeed, the second selection yielded markedly different motifs with higher and therefore more confident Weeder scores (Fig 2E), indicating that our modified protocol improves the current bind-n-seq approach.

### Selection of RNA motifs of immunoprecipitated RBPs

To establish a versatile and broadly applicable RNA motif enrichment method, we assessed different pull-down strategies and compared selection efficiencies (Fig 3A). In addition, results were

validated with an independent filter binding assay using untagged proteins (Fig 3B). We expressed the well-characterized FLAG-HA(FH)-tagged RBPs CELF1, hnRNPA1, and GRSF1 in HEK 293 cells, isolated them by anti-FLAG immunoprecipitation and used the beads-bound proteins for RNA bind-n-seq (Fig 3C). Furthermore, we immunoprecipitated endogenous proteins using RBP-specific antibodies and performed again RNA motif selection on beads (Fig 3D). Immobilization using the two different immunoprecipitation strategies, followed by selection from the 8-mer RNA library yielded a strong enrichment of UAGUGUA-containing motifs for hnRNPA1 (Fig 3C and D). Interestingly, the motif differs slightly from the one we found with recombinant proteins either in filter binding assays (Fig 3B) or bind-n-seq (Fig 1B). This might suggest a slightly different binding activity, when full-length proteins are isolated from endogenous sources. For CELF1 and GRSF1, retrieved motives were highly similar between the used methods (filter binding, recombinant proteins or beads-bound endogenous proteins, Fig 3B–D. See Fig S4 for top Weeder scores of CELF1). Our data demonstrate that endogenous proteins can be used for RNA bind-n-seq, and therefore, we refer to this improved approach as endo-bind-n-seq.

RBPs frequently exhibit tissue- or developmental stage-specific expression patterns and it would be desirable to assess their binding specificity in their dynamic natural environment. Protein modifications, for example, might differ between tissues and immortalized cell lines. Moreover, RBPs could be incorporated into different RNPs, which might also affect target recognition. To test endo-bind-n-seq in specific tissue lysates, we immunoprecipitated endogenous hnRNPA1 from mouse brain and liver samples (Fig 3E). We isolated endogenous hnRNPA1 by specific antibodies and validated the immunoprecipitate by Western blotting (data not shown). Beads-bound hnRNPA1 was used for endo-bind-n-seq (Fig 3E). To further test for specificity of our method, we used beads only as negative control. Strikingly, the known RNA motif was readily enriched from both tissue extracts, whereas the negative control did not enrich for a specific sequence element. Our data therefore demonstrate that RNA endo-bind-n-seq can be used to identify target RNA motifs of endogenous RBPs isolated from tissue samples.

### Assessing complex RNA binding activity

We next examined, whether we can also select known but more complex RNA motifs using endo-bind-n-seq. We investigated FH-Pumilio1, an RBP known to bind an extended RNA motif beyond an 8-mer (Wang et al, 2002) and therefore applied the 14-mer RNA library. Indeed, after anti-FLAG immunoprecipitation, a sequence resembling its known binding motif was selected from the RNA library (Fig 4A). Therefore, for RBPs recognizing short RNA motifs, the use of the 8-mer random library might be optimal whereas RNA

---

recombinant protein constructs are indicated below the schematically depicted domain organization of the respective proteins (bottom). **(C)** Sequence logo representation of enriched motifs after two selection rounds using 8-mer or 14-mer RNA input pools with GST- and His₆-tagged hnRNPA1(aa 2–187) constructs, encompassing its two C-terminal RRM domains. The differently tagged hnRNPA1(aa 2–187) proteins were immobilized by indicated strategies. Enrichment scores were calculated with Weeder2. The protein truncation is indicated below depicted domain organization of hnRNPA1 (right).

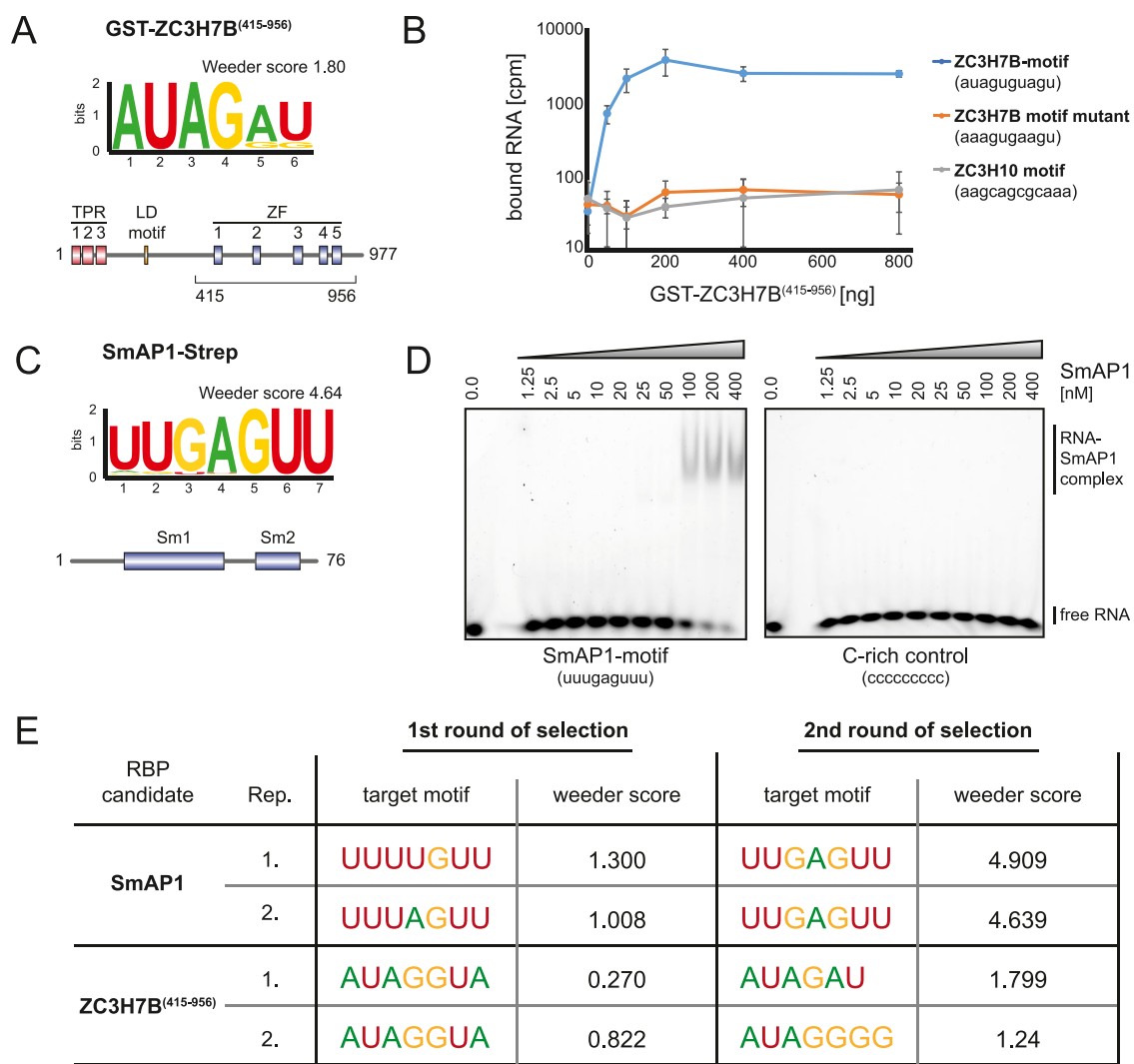

**Figure 2. Endo-bind-n-seq with uncharacterized RBP candidates.**
**(A)** Sequence logo representation of the motif enriched from 8-mer RNA input pool with purified, immobilized GST-ZC3H7B(aa 415–956) after two selection rounds in endo-bind-n-seq. Enrichment score was calculated with Weeder2. The used recombinant protein construct is indicated below the schematically depicted domain organization of ZC3H7B. **(B)** Filter binding assay with 5′ radiolabeled RNA containing the ZC3H7B-enriched motif, a mutant motif, or an unrelated control sequence (ZC3H10 motif). Retained radioactivity on the nitrocellulose filter after incubation with increasing amounts of GST-ZC3H7B(aa 415–956) is plotted against the applied amount of protein. **(C)** Sequence logo representation of the motif enriched from 8-mer input RNAs with purified, immobilized *Pfu*-SmAP1-Strep after two selection rounds in endo-bind-n-seq. The enrichment score from Weeder2 analysis and schematic domain organization of archaeal SmAP1 are shown. **(D)** Electromobility shift assay using HEX-labeled RNAs containing either the identified SmAP1-motif or a C-rich control RNA in binding reactions with increasing amounts of recombinant SmAP1 (ranging from 0 to 400 nM). Complexes were separated by native gel electrophoresis. **(E)** RNA motifs enriched by purified SmAP1-Strep and GST-ZC3H7B(aa 415–956) after the first and the second round of target RNA selection from randomized 8-mer RNA input pools in endo-bind-n-seq. Motifs and enrichment scores after Weeder2 analysis are given for two independent experiments, respectively.

14-mers are suitable for longer target sites or more complex interaction sites such as short local, structural elements.

To explore another potential endo-bind-n-seq application, we tested whether indirect binding motifs in the context of a multi-subunit RNP complex can be identified as well. We expressed FH-CSTF1, which forms a complex with CSTF2 and 3 and is involved in mRNA polyadenylation (Takagaki & Manley, 2000; Grozdanov et al, 2018). CSTF2 contains two RRMs that bind U-/GU-rich motifs (Takagaki & Manley, 1997). Strikingly, indirect isolation of CSTF2 through CSTF1 allowed for a robust selection of a GU-rich binding motif suggesting that endo-bind-n-seq can be used to study RNA binding of larger protein assemblies (Fig 4B). In contrast, recombinant CSTF1 alone did not enrich for its specific target RNA motif in endo-bind-n-seq, confirming that complex formation with CSTF2 was essential for target recognition (data not shown).

We identified a specific RNA motif for the so far uncharacterized RBP ZC3H7B (AUAGAU), when a recombinant fragment containing the five Zn fingers was used (Fig 2C). Based on various experiments (data not shown), we assumed that ZC3H7B is a weaker RNA binder, and therefore we examined, whether binding activity is different in the context of the full-length protein when isolated from natural sources. We immunoprecipitated FH-ZC3H7B from transfected HEK

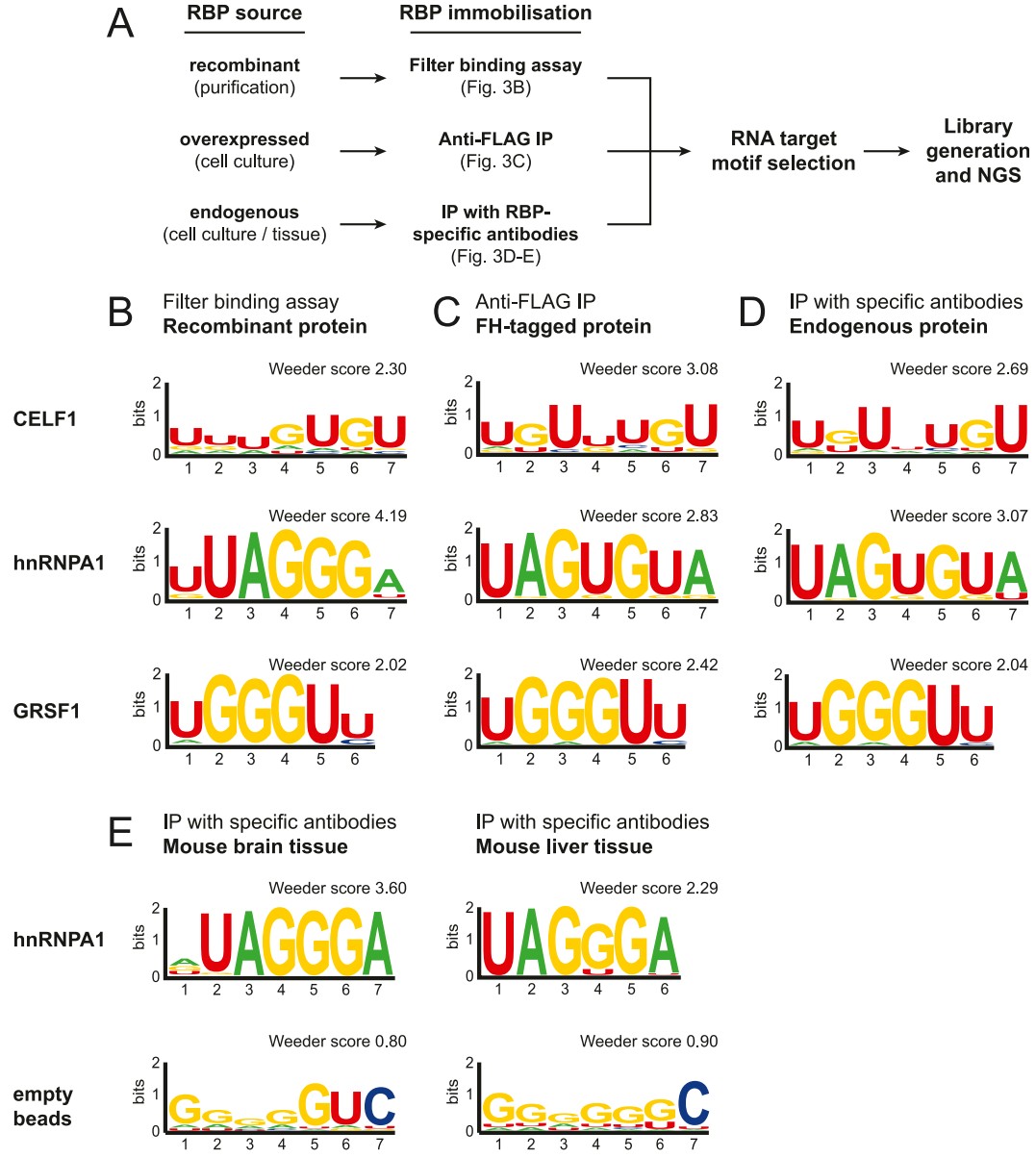

**Figure 3. Endo-bind-n-seq with RBPs originating from various sources.**
**(A)** Outline of the endo-bind-n-seq experiments conducted with RBPs from different origin. **(B, C, D)** Sequence logo representation of enriched motifs after two selection rounds in endo-bind-n-seq tested for the RBP candidates CELF1, hnRNPA1, and GRSF1. **(B, C, D)** Experiments were performed with untagged, recombinant protein domains (CELF1[aa 2–486], GRSF1[aa 134–480], hnRNPA1[aa 2–320]) immobilized in filter binding assays (B), with overexpressed full-length proteins fused to a FLAG-HA-tag (FH) after immunoprecipitation from cell lysate (C), and with endogenous proteins after immunoprecipitation with specific antibodies from cell lysate (D). Indicated enrichment scores were calculated with Weeder2. **(E)** Sequence logo representation of enriched motifs after two selection rounds with immobilized, endogenous hnRNPA1 protein from murine brain and liver lysates (top) or lysate-incubated, empty beads as negative control. Loaded and empty beads were incubated with 8-mer RNA input pool. Enrichment scores were calculated by Weeder2.

293 cells and performed endo-bind-n-seq (Fig 4C). Surprisingly, we enriched an RNA motif that differs from the one that was identified with the recombinant Zn finger domain, suggesting altered binding specificity. To further analyze this assumption, we performed competition experiments (Fig 4D and E). FH-ZC3H7B was immunoprecipitated and incubated with the radiolabeled RNAs containing either the AUAGAU or the AGUUUCG motif to allow for binding. Samples were subsequently incubated with non-labeled competitor RNA containing the other motif. Interestingly, both RNAs were efficiently bound in these assays, suggesting broader and more complex RNA binding activity. Furthermore, the unlabeled RNA motif identified with the recombinant Zn fingers competed better, which was rather unexpected and still remains puzzling. These results may suggest that different motifs are selected under different experimental conditions, or under different protein or RNA concentrations. Our data further underscore that not all RNA binding activities can easily be assessed using in vitro selection methods and particularly low affinity binders may generate motifs

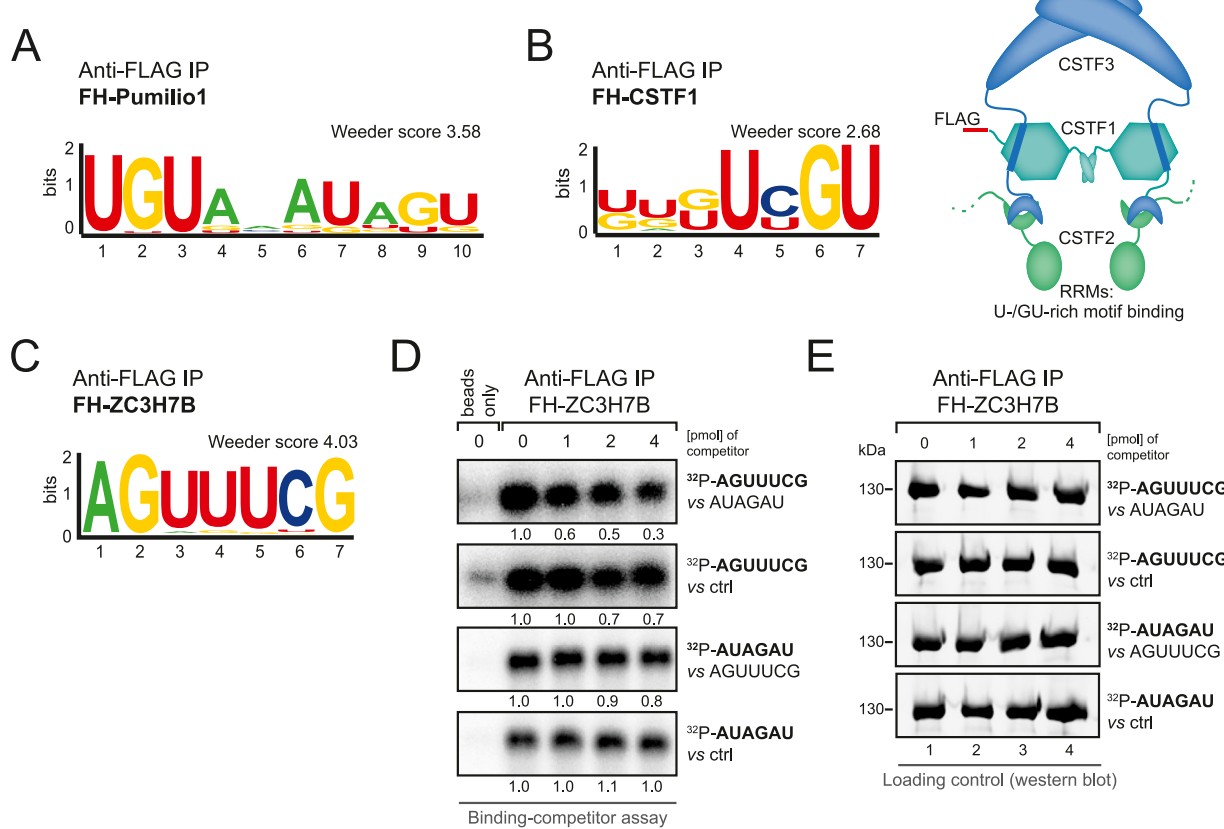

**Figure 4. Endo-bind-n-seq in context of more complex RNA binding functions.**
**(A)** Sequence logo representation of the motif enriched by overexpressed FH-Pumilio1 after two selection rounds in endo-bind-n-seq with a randomized 14-mer RNA input pool. Resulting enrichment score from Weeder2 analysis is given. **(B)** Sequence logo representation of the enriched motif after two selection rounds with immobilized FH-CSTF1 and incubation with randomized 8-mer RNA input pool. Indicated enrichment score was calculated with Weeder2 (left). Subunit organization of the cleavage stimulation factor (CstF) complex is schematically depicted (right). **(C)** Sequence logo representation of the enriched motif from 8-mer RNA input pool after two selection rounds with overexpressed FH-ZC3H7B. Enrichment score was calculated with Weeder2. **(D)** Radioactive competitor assay with RNAs containing either the motif enriched in endo-bind-n-seq by purified GST-ZC3H7B(aa 415–956) (AUAGAU) or by overexpressed FH-ZC3H7B (AGUUUCG), and an unrelated control sequence (AGAGAG). Phosphor image of binding reactions with fixed amounts of labeled RNA (1 pmol) and competition with indicated increasing amounts of unlabeled competitor RNA. **(E)** Western blot analysis of immunoprecipitated FH-ZC3H7B samples used in the RNA competitor assay.

with low confidence that need to be thoroughly validated in in-dependent in vivo experiments.

## Protein amounts affect RNA motif selection

Our observation that motifs can vary depending on the protein (full-length versus RBD) or the used method (recombinant protein vs. immunoprecipitated proteins) suggests, that selection conditions have to be considered carefully. To better understand selection conditions, we first assessed the influence of protein levels on motif selection. RBPs bind to their target RNAs in the context of distinct local concentrations, which is the basis for their RNA affinity. If protein concentrations are too high, unspecific interactions might occur. If they are too low, off-rates might dominate the binding kinetics and motif selection becomes inaccurate and weak. To study concentration effects of immunoprecipitated samples, which are more difficult to quantify than recombinant proteins, we performed protein titration experiments and generated a reference curve (Fig 5A and B). We expressed GST- and HA-

tagged hnRNPA1[2–187], purified it and analyzed increasing amounts by HA-tag-specific Western blotting (Fig 5A). Quantified western blot signals were fitted to a reference curve using the formula presented in Fig 5B (see the Materials and Methods section for details). Since we use the HA-specific antibody for our subsequent endo-bind-n-seq experiments, the curve serves as a quantification standard for protein amounts in the individual samples. As testing cases, we chose hnRNPA1 as a strong and robust RNA binder and ZC3H7B, which is most likely a weak RNA binder that had enriched for different motifs in our previous experiments. To estimate protein amounts immunoprecipitated from lysates, we expressed FH-ZC3H7B and FH-hnRNPA1 in HEK 293 cells and analyzed anti-FLAG immunoprecipitates using increasing amounts of total lysate by Western blotting (Fig 5C, left panel). Using our standard curve, we estimated the concentrations of the FH-tagged proteins in our endo-bind-n-seq reactions (Fig 5C, right panel). With the assumption that beads are not saturated, amounts of total lysates can now be easily adjusted to reach a desired protein level for endo-bind-n-seq experiments. We used decreasing amounts of FH-

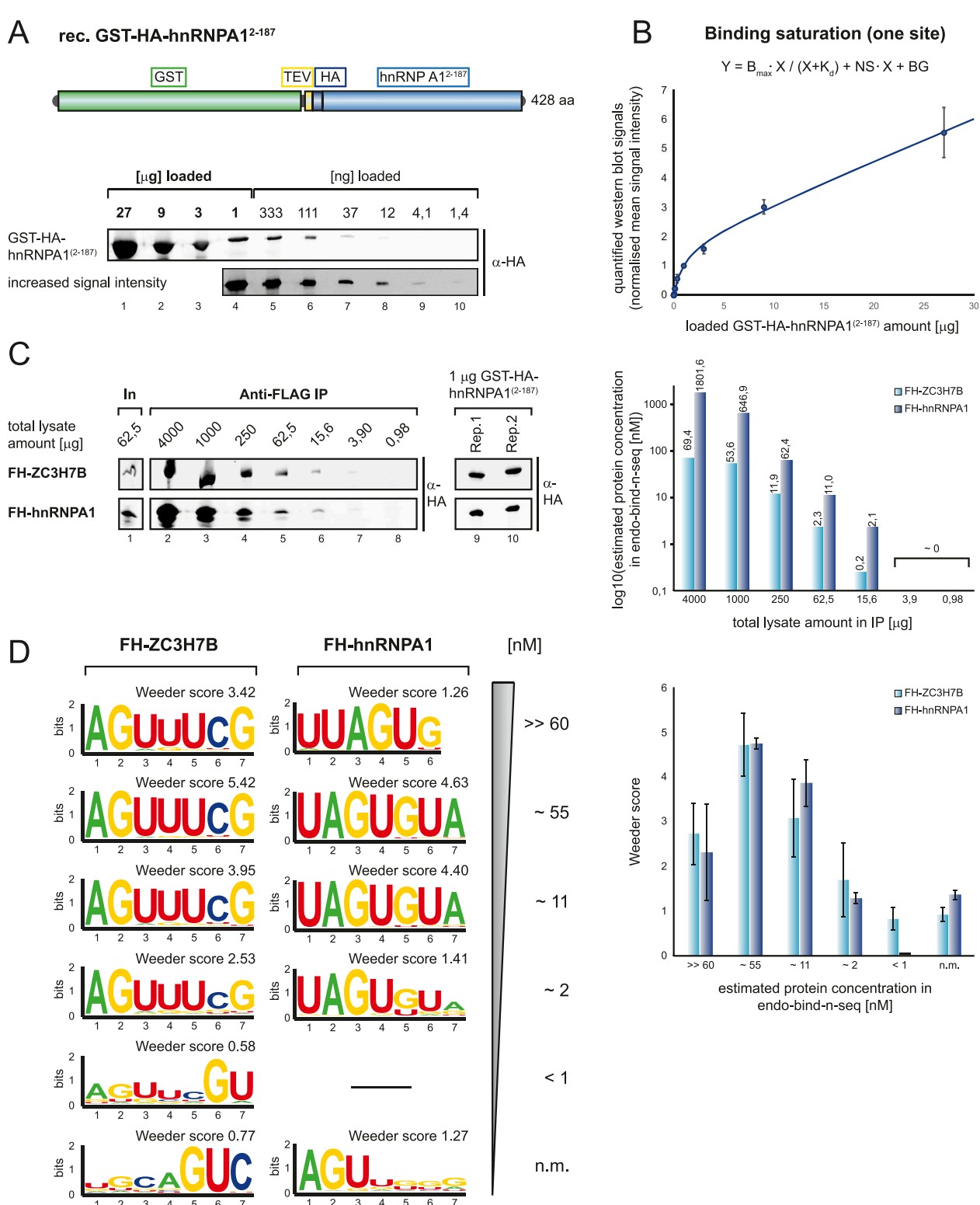

**Figure 5. Quantification of optimal protein concentrations in endo-bind-n-seq.**
**(A)** Schematically depicted domain organization of the used reference protein construct GST-HA-hnRNPA1(2–187) (top). Defined amounts of purified GST-HA-hnRNPA1(2–187) were titrated and detected by western blot (bottom). The incorporated HA-tag (dark blue) served as immunodetection site in the reference protein construct. **(B)** Western blot signal intensities detected from GST-HA-hnRNPA1(2–187) titration blots were quantified and normalized to immunosignals detected with 1 μg of loaded reference protein. Mean intensity signals from four independent replicates were plotted against the respectively loaded protein amounts. Corresponding regression curve was analyzed according to a one site dependent binding saturation, applying the indicated equation (Bmax: maximum specific binding; Kd: equilibrium dissociation constant; NS: nonspecific binding; BG: background; defined by Goodness-of-Fit approximation). **(C)** Representative Western blots of anti-FLAG IP experiments

hnRNPA1 and FH-ZC3H7B and performed endo-bind-n-seq (Fig 5D). In both cases, specific RNA motifs could be found enriched when estimated concentrations above 1 nM were applied (Fig 5D, left panel). Moreover, when blotting the individual Weeder motif scores against the estimated protein concentration, we found a peak around 55 nM (Fig 5D, right panel). Our data therefore suggest a concentration optimum for RBPs in endo-bind-n-seq. Above and below this optimum, the detected binding motifs become statistically less significant. Rather expectedly, very low protein concentrations result in background signals with no relevant motifs.

### Influence of buffer conditions and salt concentrations

RBPs use a large variety of physical interactions to bind their individual targets. Consequently, buffer conditions and particularly salt concentrations influence RNA binding activity. To assess the extent of such effects and to further optimize endo-bind-n-seq, we performed a panel of experiments using different buffer conditions. First, we rationalized that washing stringency during protein isolation may influence the significance of endo-bind-n-seq because impurities, such as pre-bound RNA or proteins, are removed. We used increasing salt concentrations during the immunoprecipitation of the RBP and performed endo-bind-n-seq under low salt conditions (Fig 6A). FH-hnRNPA1 or FH-ZC3H7B was expressed in HEK 293 cells and immunoprecipitated using anti-FLAG-coupled beads. As negative control, empty beads were incubated with WT HEK 293 lysate. Immunoprecipitations were performed and washed under the indicated salt concentrations. RIPA buffer additionally contained increased amounts of detergent. Western blot analysis confirmed comparable amounts of bound proteins (Fig 6B). For FH-hnRNPA1, the core motif UAGU/GG was readily detected under all salt conditions with similar Weeder enrichment scores (Fig 6A and C), suggesting that high-affinity interactions are less prone to background in endo-bind-n-seq. For FH-ZC3H7B, the core sequence AGUUU was detected under 300 mM, 500 mM, and 1 M salt conditions with Weeder scores peaking at 300 and 500 mM (Fig 6A and C). The motif, however, became less clear when low salt or RIPA buffer was used, suggesting that proteins with lower RNA affinities might be more sensitive to immunoprecipitation conditions. Of note, empty beads also enriched distinct motifs (Fig 6A), and therefore, it is advisable to include such controls in endo-bind-n-seq experiments.

In a second approach, we used low salt conditions for immunoprecipitation and washing but performed the actual RNA selection under increasing salt concentrations, which directly affects RNA-protein interaction (Fig 6D). FH-hnRNPA1 was isolated and used for endo-bind-n-seq. Interestingly, the core motif UAGU/GG was only found when 300, 500 mM, or 1 M salt was used. Weeder enrichment scores peaked at 500 mM, suggesting optimal binding conditions for hnRNPA1 (Fig 6D and E). The binding reaction in 150 mM salt buffer resulted in a motif indistinguishable from the

background control (Fig 6D and E). Taken together, these data suggest that buffer conditions are an important component of endo-bind-n-seq and should be considered for each individual RBP that is investigated.

# Discussion

Post-transcriptional gene regulation is facilitated by various means. Central to this process, however, are RBPs, which interact with distinct target RNAs and may directly affect RNA half-lives or translation, for example. To understand such regulatory processes, detailed knowledge about the RNA binding activity of RBPs is essential and therefore methods have been developed to identify RNA motifs that RBPs bind. Among others, these approaches include in vitro studies such as RNAcompete or RNA bind-n-seq or in vivo methods such as many CLIP variants. Joined efforts of many labs yielded in a comprehensive overview of the RNA binding landscape as well as RNA binding affinities of individual RBPs (Ray et al, 2013; Van Nostrand et al, 2020; Tripto & Orenstein, 2021; Jouravleva et al, 2022). Although all these methods proved to be highly valuable and robust, they also have limitations. CLIP experiments, for example, are meant to "freeze" the binding landscape in vivo by UV-crosslinking protein-RNA interactions. However, affinities are difficult to determine and the vast number of binding sites that are typically reported suggest that transient and potentially non-productive interactions might be captured as well. How many of such sites are physiologically meaningful remains unclear and requires rigorous validation. In vitro approaches can accurately provide Kd values and a very solid biochemical understanding of individual protein-RNA interactions. However, if and how identified target sites are bound in a physiological context needs to be separately assessed by other methods. Furthermore, in vitro approaches rely on the availability of recombinant, correctly folded and functional RBPs, which is not always possible. To overcome this technical limitation and to provide a tool that can be applied also in non-biochemistry labs, we have developed endo-bind-n-seq. We isolate RBPs from their physiological environments and perform on-beads selection of RNA motifs. Our protocol is robust and lysates from cell lines or primary tissues can easily be used for immunoprecipitations. However, when we optimized our protocol, we encountered several difficulties and unexpected aspects. First, it is important that optimal RBP concentrations are used and this might differ for each RBP. Low-affinity binders may require higher concentrations than high affinity binders. Moreover, when concentrations are too high, unspecific binding occurs and motifs become less clear. Second, buffer and salt conditions need to be optimized for each individual RBP and thus an overarching protocol, that would work for all RBPs, could hardly be developed.

---

on titrated, defined amounts of overexpression cell lysates, containing either FH-ZC3H7B or FH-hnRNPA1 protein (left). 1 µg of GST-HA-hnRNPA1(2–187) samples serve as internal reference for signal intensity normalization. Immunoprecipitated protein amounts were analyzed from three independent replicates, applying the quantification system of Fig 5B. Resulting calculated final protein concentrations in endo-bind-n-seq reactions are plotted against the respectively applied lysate amounts (right). **(D)** Sequence logo representation of enriched motifs after two selection rounds with 8-mer RNA input pool, testing defined concentrations of FH-ZC3H7B and FH-hnRNPA1 in endo-bind-n-seq following anti-FLAG IP. Quantified, estimated protein concentrations are indicated. Enrichment scores were calculated with Weeder2 (left). Mean enrichment scores from Weeder2 analysis of two independent experiments are plotted against tested protein concentrations (right).

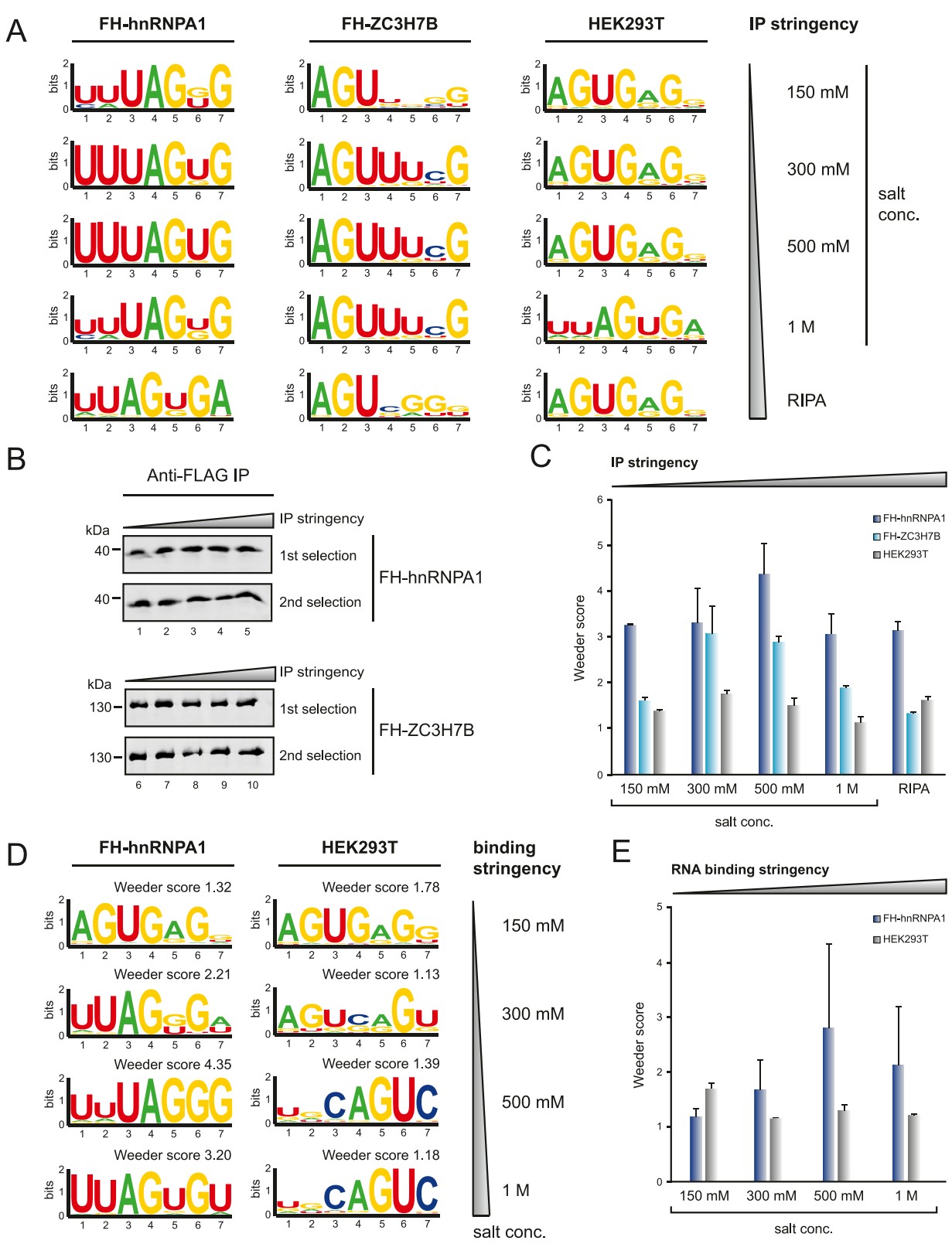

**Figure 6. Optimization of buffer conditions for protein immobilization and RNA binding.**
**(A)** Sequence logo representation of enriched motifs after two selection rounds with immobilized FH-hnRNPA1 (left), FH-ZC3H7B (middle), or with HEK293T WT lysate incubated, empty FLAG-beads (right). After lysate incubation, beads were washed with indicated buffer stringency conditions. Stringency increases from top to bottom by salt concentration and ends with RIPA buffer containing high amounts of detergent. **(B)** Western blot analysis of immunoprecipitated FH-hnRNPA1 and FH-ZC3H7B after IP washes with increasing buffer stringency conditions. Representative blots for samples of the first and second selection round. **(C)** Mean enrichment scores of highest

Instead, each RBP requires protocol validation, particularly since relatively robust motifs can be enriched with beads-only negative controls when nonoptimal conditions are used.

Another limitation of in vitro assays using recombinant proteins is that RBPs can be post-translationally modified in cells and this might influence target RNA binding. For example, PUM1 has 17 annotated modification sites, of which one phosphorylation (Ser714) was reported to promote RNA binding (Kedde et al, 2010). Whereas in our hands, recombinant PUM1-RBD (aa 828–1,176) did not work properly in bind-n-seq experiments (data not shown), the immunoprecipitated protein selected the known binding site UGUANAU with good enrichment scores (Fig 4A). It is tempting to speculate that PUM1 isolated from cells is phosphorylated, thus allowing for RNA binding. It is also possible that RBPs may bind to different sequences based on the post-translational modification status, which can also be monitored by endo-bind-n-seq.

Endo-bind-n-seq further allows for RNA motif selection of protein complexes. Proteins that indirectly interact with RNA through an RBP binding partner can be immunoprecipitated and the co-isolated RBP may yield an RNA motif during selection. Furthermore, even RBPs can be in complex with other RBPs and thus more complex motifs might be selected. Here, the 14-mer library would be the RNA pool of choice for the endo-bind-n-seq selection rounds.

Nonetheless, endo-bind-n-seq may also have limitations. Low affinity binding sites could be difficult to select. Or, RNPs with multiple RBDs may interact with different motifs and resulting data analysis is different because several motifs overlap in the sequencing dataset. In such cases, not only the most enriched motif but also less-enriched motifs should be considered and tested further. This might also explain the variation of motifs bound by ZC3H7B that we observed in our assays. Taken together, endo-bind-n-seq provides a valuable extension of bind-n-seq, adding several features that were not covered by other methods so far.

## Materials and Methods

### Randomized RNA libraries

8- and 14-mer RNA libraries were purchased from Metabion using random insertion of each of the four nucleotides during chemical synthesis. Sequence biases, that were discovered after sequencing, were removed by adjusting nucleotide concentrations during synthesis (see Table S1).

### Binding reaction with GST-tagged, recombinant protein

20 $\mu$l of *Glutathione Sepharose* (GE healthcare) were washed twice with binding buffer (25 mM Tris–HCl pH 7.5, 150 mM KCl, 3 mM MgCl$_2$, 0.01% [vol/vol] NP-40, 1 mg/ml BSA, 1 mM DTT, 15 $\mu$g/ml heparin, 5% [vol/vol]

glycerol) and resuspended in 400 $\mu$l of the binding buffer, containing 40 U *RiboLock* (Thermo Fisher Scientific) and 10 $\mu$g of randomized, phosphorylated 8-mer RNA pool or 15 $\mu$g of the 14-mer RNA pool. GST fusion protein was added to a final concentration of 100 nM, and the binding reaction was performed for 30 min at RT with agitation. Beads were collected by centrifugation (1,000$g$, 2 min, 4°C) and washed three times with ice-cold binding buffer. After the last washing step, all residual washing buffer was carefully removed before adding 200 $\mu$l elution buffer (10 mM Tris–Cl pH 7.0, 400 mM NaCl, 1 mM EDTA, 1% [wt/vol] SDS). Elution was performed at 55°C in a ThermoMixer C device (Eppendorf), shaking at 700 rpm for 5 min before RNA extraction.

### Binding reaction with His-tagged recombinant protein

Selection reactions with His$_6$-tagged protein were carried out as described for GST-tagged variants, with the exception that IMAC-Sepharose beads (GE healthcare) charged with Nickel chloride were used as immobilization matrix and DTT in the binding buffer was omitted.

### Binding reaction with Strep-tagged recombinant protein

Selection reactions with Strep-tagged protein were carried out as described for GST-tagged variants, with the exception that MagStrep "type3" XT beads (IBA Lifesciences) were used as immobilization matrix.

### Binding reaction with Epitope-tagged recombinant protein

For immobilization of recombinant HA-tagged protein, 2 $\mu$g of monoclonal anti-HA antibody (Covance) were incubated with 20 $\mu$l Protein G Sepharose Beads (GE Healthcare) in 300 $\mu$l IP wash buffer (50 mM Tris–HCl pH 7.5, 150 mM NaCl, 0.05% [vol/vol] NP 40, 1 mM DTT) for 1 h at 4°C. The beads were washed two times with binding buffer and then used in a binding reaction as described for GST-tagged, recombinant proteins, now with the immobilized Epitope-tagged protein.

### Binding reaction with untagged, recombinant proteins—filter binding reaction

For untagged recombinant proteins, a filter binding assay was used for the immobilization and identification of selected RNAs. For this, 100 nM recombinant protein in 400 $\mu$l filter binding buffer (25 mM Tris–HCl pH 8.0, 50 mM KCl, 2 mM MgCl$_2$, 0.01% NP-40, 1 mM DTT, 10% [vol/vol] glycerol) containing 40 U *RiboLock* (Thermo Fisher Scientific) was mixed with 10 or 15 $\mu$g phosphorylated 8-mer, or 14-mer RNA pool and incubated for 30 min at RT whereas rotating. Nitrocellulose membrane (GE Healthcare) was equilibrated with filter binding buffer containing 15 $\mu$g/ml heparin and fitted to a glass drip directly before spotting the binding reaction to the membrane under vacuum. The membrane was washed with 20 ml ice-cold filter binding buffer. The spot harboring the binding reaction was cut out

---

scoring motifs after Weeder2 analysis from two independent experiments, plotted against the applied IP stringency condition in endo-bind-n-seq with FH-hnRNPA1 (dark blue), FH-ZC3H7B (light blue) and HEK293T wild-type (gray). **(D)** Sequence logo representation of enriched motifs after two selection rounds with immobilized FH-hnRNPA1 and empty FLAG-beads, incubated with wild-type HEK293T lysate. RNA binding buffer stringencies increase by indicated salt concentrations from top to bottom. **(E)** Mean enrichment scores of highest scoring motifs from Weeder2 analysis of two independent experiments, plotted against increasing RNA binding stringency conditions applied in endo-bind-n-seq with immobilized FH-hnRNPA1 (light blue), or HEK293T wild-type lysate incubated FLAG-beads (gray).

and transferred to a 1.5 ml reaction tube. The RNA was eluted from the membrane by adding 250 µl NA-45 buffer (50% formamide, 1.8 M sodium acetate, 2 mM EDTA, 0.2% [wt/vol] SDS) and shaking at 70°C for 30 min. The buffer solution was then transferred to a fresh reaction tube for subsequent RNA extraction.

### Binding reaction with immunoprecipitated FLAG-HA-tagged protein

HEK293T cells were transfected by the calcium-phosphate method with a plasmid containing the coding sequence of a candidate RBP, fused to an N-terminal FLAG-HA-tag under control of a CMV promotor for high expression. Cells were grown for 1.5–2 d after transfection and collected by scraping the cells after washing twice with PBS. For each protein, defined lysate amounts (or without previous quantification: two 15 cm cell culturing dishes) were used per selection reaction. The cells were collected by centrifugation (5 min, 500$g$, 4°C) and resuspended in 1 ml IP lysis buffer (50 mM Tris–HCl, pH 7.5, 300 mM NaCl, 1 mM AEBFS, 1 mM DTT, 0.5% [vol/vol] NP-40). Cells were lysed by incubation on ice for 15 min. Insoluble material was pelleted by centrifugation (20,000$g$, 4°C, 15 min) and the supernatant was transferred to a fresh reaction tube. 20 µl FLAG-M2 Agarose Beads (Sigma-Aldrich) were washed twice with IP wash buffer (50 mM Tris–HCl, pH 7.5, 300 mM NaCl, 0.05% [vol/vol] NP-40), resuspended in 100 µl IP lysis buffer and added to the lysate. The binding reaction was incubated at 4°C for 2–3 h while agitating. The beads were subsequently washed three times with 1 ml IP wash buffer. During the third wash, the beads were transferred to a fresh tube and 5% of the beads were taken for Western blot analysis of the immunoprecipitation. The immunoprecipitated proteins were directly applied in an RNA-binding reaction by resuspending the beads in 400 µl binding buffer and addition of randomized RNA input pool (usually 8-mer, if not differently annotated). Binding reactions were further conducted as described for GST-tagged, recombinant proteins. In case of stringency titration experiments, all steps were performed as described above. However, buffers were varied according to the list below.

**IP wash buffer (IP stringency titration).**

|  |  |
| --- | --- |
| Base composition (IP-WB) | 50 mM Tris–HCl pH 7.5, 0.05% (vol/vol) NP-40, 1 mM DTT |
| NaCl concentration | IP-WB (150 mM); IP-WB2 (300 mM); IP-WB3 (500 mM); IP-WB4 (1 M); RIPA buffer (150 mM, *with following base composition*: 50 mM Tris–HCl pH 7.5, 1.0% [vol/vol] sodium deoxycholate, 0.1% [vol/vol] SDS, 1.0% [vol/vol] NP-40, 1 mM DTT) |

**RNA binding buffer (RNA binding stringency titration).**

|  |  |
| --- | --- |
| Base composition (BB) | 25 mM Tris–HCl pH 7.5, 3 mM MgCl$_2$, 0.01% (vol/vol) NP-40, 1 mg/ml BSA, 1 mM DTT, 5% (vol/vol) glycerol |
| KCl concentration | Low-salt BB (75 mM); BB (150 mM); High-salt BB (300 mM); High-salt BB2 (500 mM); High-salt BB3 (1 M) |

### Binding reaction with endogenous proteins from cell lysates

HEK293T cells were grown to 100% confluency and harvested by scraping the cells after washing twice with PBS. For each protein, defined lysate amounts (or without previous quantification: four 15 cm cell culturing dishes) were used per binding reaction. The cells were collected by centrifugation (5 min, 500$g$, 4°C) and resuspended in 2 ml IP lysis buffer (50 mM Tris–HCl, pH 7.5, 300 mM KCl, 1 mM AEBFS, 1 mM DTT, 0.5% [vol/vol] NP-40). The cells were lysed for 15 min on ice. Insoluble material was pelleted by a centrifugation step (20,000$g$, 4°C, 15 min) and the supernatant transferred to a fresh reaction tube. To couple specific antibodies to beads, 20 µl *protein-G Sepharose* (GE Healthcare) were washed twice with IP wash buffer (50 mM Tris–HCl, pH 7.5, 300 mM KCl, 0.05% [vol/vol] NP40) and resuspended in 300 µl IP lysis buffer. 2 µg of the specific antibody were added to the beads and coupled for 1 h at 4°C whereas rotating. Beads were washed twice with 1 ml IP wash buffer, resuspended in 100 µl IP lysis buffer and added to the cell lysate. The immunoprecipitation and RNA binding reactions were performed analogous to the FLAG-HA-tagged proteins. Used antibodies for protein immunoprecipitation are listed below.

**Antibodies used for immunoprecipitation of endogenous protein candidates.**

| Antibody | | Source |
| --- | --- | --- |
| Anti-HA | HA.11, Clone 16B12; Covance | mouse, mAb |
| Anti-HNRNPA1 | sc-32301; Santa Cruz | mouse, mAb |
| Anti-CELF1 | 13002-1-AP; Proteintech | rabbit, pAb |
| Anti-GRSF1 | ERP16678, ab205531; Abcam | rabbit, mAb |

### Binding reaction with endogenous proteins from murine tissue

Brain and liver tissue samples were mechanically homogenized in 1 ml IP lysis buffer (50 mM Tris–HCl, pH 7.5, 300 mM KCl, 1 mM AEBFS, 1 mM DTT, 0.5% [vol/vol] NP-40) by mortar and pestle, with 2–3 shock freezing steps with liquid nitrogen added during grinding. The lysed tissue was transferred into reaction tubes and insoluble material was pelleted by centrifugation (20,000$g$, 4°C, 30 min). Immunoprecipitation was performed from supernatants with specific antibodies coupled to protein-G Sepharose as described above for endogenous proteins from cell lysates.

### RNA extraction from target RNA selection reactions

250 µl Roti Aqua PCI (for RNA extraction, Carl Roth) were added to the elution reaction and mixed thoroughly by vortex. After centrifugation (17,000$g$, 10 min, 20°C) for phase separation, the aqueous phase was transferred into a new reaction tube. 20 µg Glycogen (RNA grade; Thermo Fisher Scientific) and 0.8 ml ethanol were added, the reaction well mixed and kept at −20°C for at least 1 h. RNA was pelleted by centrifugation (20,000$g$, 30 min, 4°C), the supernatant was completely removed, and the pellet was dried for 5 min at 55°C.

## Library preparation

**First adapter ligation** Extracted RNA pellet was resuspended in 12 $\mu$l RNase-free water, shaking 5 min at 55°C, and then placed on ice. 2 $\mu$l 10x ligation buffer (500 mM Tris–HCl pH 7.5, 100 mM MgCl$_2$, 10 mM DTT), 3 $\mu$l DMSO and 1 $\mu$l pre-adenylated 3′-adapter were added (5′-rApp-AAACTGGAATTCTCGGGTGCCAAGG-ddC-3′), heated for 30 s at 95°C and immediately placed on ice. The reaction was supplemented with 20 U *RiboLock* (Thermo Fisher Scientific) and 2 $\mu$l truncated RNA ligase 2 (RNL2 [aa 1–249] K227Q), mixed and incubated overnight at 4°C. Ligation was completed by further incubation at 37°C for 1 h and subsequently placed on ice.

**Second adapter ligation** 2 $\mu$l 10x ligation buffer, 3 $\mu$l DMSO, 0.4 $\mu$l ATP (100 mM), 1 $\mu$l 5′-RNA adapter oligo (10 $\mu$M; 5′-GUUCA-GUAAUACGACUCACUAUAGGG-3′), and 11.6 $\mu$l RNase-free water were added to the first ligation reaction. The mix was heated to 95°C for 30 s and directly placed on ice. 20 U *T4 RNA ligase 1* (NEB) were added, mixed, and incubated at 37°C for 1 h.

**cDNA synthesis** For cDNA synthesis, the *First Strand cDNA Synthesis Kit* (Thermo Fisher Scientific) was used. 1 $\mu$l RT-primer (20 $\mu$M; 5′-GCCTTGGCACCCGAGAATTCCAGTTT-3′) was mixed with 10 $\mu$l of the second adapter ligation reaction and annealed for 5 min at 65°C. After cooling briefly on ice, 4 $\mu$l reaction buffer, 2 $\mu$l dNTPs (10 mM), 1 $\mu$l *RiboLock*, and 2 $\mu$l MuLV-RT were added and cDNA-synthesis performed for 1 h at 37°C, according to manufacturer's instructions.

**Barcoding PCR** The synthesized cDNA was used in a PCR to introduce barcoding primer for next-generation sequencing of RBP selected RNAs. For this, 5 $\mu$l of cDNA reaction were mixed with 1 $\mu$l 3′ barcode primer (Trueseq System, 100 $\mu$M), 1 $\mu$l 5′ barcode primer (100 $\mu$M), 1.25 $\mu$l dNTPs (10 mM), 10 $\mu$l 5x HF buffer, 1 U *Phusion polymerase* (Thermo Fisher Scientific), and 31.25 $\mu$l water. The PCR program consisted of a 3 min denaturing step at 98°C, 30 amplification cycles (*denaturing*: 98°C, 10′′; *annealing*: 60°C, 30′′; *elongation*: 72°C, 10′′) and a final elongation step for 5 min at 72°C. Following primer sequences were used:

3′ barcode primer:

CAAGCAGAAGACGGCATACGAGAT**NNNNNN**GTGACTG GAGTTCCTTGGCACCCGAGAATTCCA.

5′ barcode primer:

AATGATACGGCGACCACCGAGATCTACAC**NNNNNN**GTTCAGTAATACGAC TCACTATAGGG.

**NOTE:** Depending on the barcode sequence composition, some 5′ barcode primers need to be further supplemented with an additional CAC-trinucleotide downstream to the barcode hexamer. With this, melting temperatures get optimized toward the sequencing primers, increasing the final output of numbers of reads.

## Urea PAGE of barcoding PCR products

A 6% urea acrylamide gel (15 × 22 cm) was pre-run in TBE-buffer at 250 V without heating to avoid denaturation of DNA samples. PCR samples were supplemented with 12 $\mu$l *6x TriTrack DNA loading dye* (Thermo Fisher Scientific) and as reference, 20 $\mu$l *Ultra Low Range DNA Ladder* (Thermo Fisher Scientific) was used. Prior loading, wells

were flushed with TBE-buffer. Gels were run at 250 V until the bromophenol blue dye had reached the lower end of the gel (4–6 h) and subsequently stained with ethidium bromide. Under UV-light, a close double-band could be observed with adapter dimers running around 140 bp, whereas ligation products containing the desired insert sequence appeared similar to the 150 bp ladder band. Insert-containing product bands were cut out, gel pieces were crushed and the DNA eluted with 300 $\mu$l of 400 $\mu$M NaCl overnight at 4°C with agitation. Gel pieces were pelleted by centrifugation (20,000$g$, 10 min, 4°C) and the supernatant containing the eluted DNA was transferred to a fresh tube. The DNA was precipitated by adding 20 $\mu$g Glycogen (Thermo Fisher Scientific) and 0.8 ml ethanol, followed by 1 h incubation at –20°C and pelleting by centrifugation (20,000$g$, 30 min, 4°C). After removal of the supernatant, the DNA was dried at 55°C for 10 min and resuspended in 30 $\mu$l water. Typical yields ranged 10–40 ng/$\mu$l of DNA. Next to further processing of the DNA for the generation of a new RNA pool, purified DNA products were applied in deep sequencing, reflecting the outcome of the first (or the second) RNA selection round in endo-bind-n-seq.

## Scale-up PCR

50 ng of the purified adapter-insert DNA product was used in a 100 $\mu$l scale-up PCR containing 1x HF buffer, 0.4 M dNTPs, 2 U Phusion polymerase (Thermo Fisher Scientific), 1 $\mu$M forward primer (5′-AATGATACGGCGACCACCGAGATCTACACGTTCAGTAATACGACTCACTA TAGG-3′), and 1 $\mu$M reverse primer (5′-GCCTTGGCACCCGAGAATTCCA GTTT-3′), following the PCR cycling program of the barcoding reaction. The PCR product was purified with the *NucleoSpin Gel and PCR Clean-up Kit* (Macherey and Nagel) and eluted with 30 $\mu$l water. Typical yields ranged between 100–150 ng/$\mu$l DNA.

## Enzymatic cleavage with MssI

3.5 $\mu$l *FastDigest* 10x buffer and 1.5 $\mu$l MssI *FastDigest* enzyme (Thermo Fisher Scientific) were added to the purified PCR product. The cleavage reaction was incubated at 37°C for 1–2 h for removal of the 3′ adapter sequence.

## In vitro transcription and gel purification of the RNA

The MssI cleavage reaction was directly applied to an in vitro transcription reaction (total volume of 500 $\mu$l), containing 30 mM Tris–HCl, pH 8.0, 25 mM MgCl$_2$, 10 mM DTT, 0.2 mM spermidine, 0.01% Triton X-100, 5 mM ATP, 5 mM CTP, 5 mM GTP, 5 mM UTP, 4 U/ml thermostable inorganic pyrophosphatase (NEB), 80 U/ml *RiboLock* (Thermo Fisher Scientific), and 0.1 mg/ml T7 RNA polymerase. The reaction was incubated at 37°C overnight.

The transcription reaction was purified on a 18% urea acrylamide gel (15 × 22 cm) after addition of 0.5 ml RNA loading buffer (formamide containing 0.005% [wt/vol] bromophenol blue). The gel was run for 6–7 h at 400–500 V, until the bromophenol blue dye reached the lower edge of the gel, and analyzed by UV shadowing. The transcribed RNA resulting from the 8-mer input pool consisted of 15 nucleotides (5′-GGGNNNNNNNNNGUUU-3′) and run about 3 cm above the bromophenol blue dye front. RNA transcribed from 14-mer input pool products contained 21 nucleotides (5′-GGGNNNNNNNNNNNNNNNNGUUU-3′) and

run about 5 cm above the dye front. The bands were excised and crushed. For elution of the RNA, 1.8 ml RNase-free water were added to the gel pieces and rotated overnight at 4°C. Gel pieces were pelleted by centrifugation (3,300$g$, 10 min, 4°C), and the supernatant was split in 2 × 650 ml into fresh reaction tubes. For precipitation of the RNA, 20 $\mu$g Glycogen (RNA grade, Thermo Fisher Scientific), 50 $\mu$l of 5 M NaCl and 600 $\mu$l 2-propanol were added to each tube and, after mixing thoroughly, the mix was stored at –20°C for at least 1 h before pelleting of precipitated RNA by centrifugation (20,000$g$, 30 min, 4°C). RNA pellets were washed once with 1 ml 80% (vol/vol) ethanol. For complete removal of residual liquid, the RNA pellet was dried at 55°C for 5 min, dissolved in 50 $\mu$l RNase-free water, and related samples were pooled. Typical concentrations of combined RNA solutions of 100 $\mu$l ranged between 80–250 ng/$\mu$l.

**NOTE:** This is the only occasion to determine the exact RNA yield. Therefore, the measurement is crucial to apply proper amounts of the transcribed RNA pool for the second round of selection in endo-bind-n-seq.

## Dephosphorylation

To remove the 5′-triphosphate from the in vitro transcription product, 11.5 $\mu$l PNK buffer A and 2 U *FastAP* (alkaline phosphatase; Thermo Fisher Scientific) were added to the RNA. The reaction was incubated for 1 h at 37°C before inactivation of the enzyme by heating at 75°C for 10 min.

## Rephosphorylation

To restore the RNA 5′-monophosphate allowing for adapter ligation, 1 $\mu$l ATP (100 mM) and 2 $\mu$l PNK (1 mg/ml) were added to the heat-inactivated dephosphorylation reaction and incubated for 1 h at 37°C. The RNA was precipitated by adding 100 $\mu$l of a 1 M NaCl solution and 0.8 ml ethanol, thorough mixing and incubation for at least 1 h at –20°C. The sample was centrifuged (20,000$g$, 30 min, 4°C) and the pellet was dried for a few minutes at 55°C and dissolved in 30 $\mu$l RNase-free water. Since ATP was coprecipitated, a photometric concentration determination of the RNA was not possible at that point. Therefore, final concentrations were calculated based on the measurements after gel purification of the RNA. The newly generated RNA

pool was then applied for a second round of target motif selection in endo-bind-n-seq.

### Library analysis

**Deep sequencing and output processing** Generated endo-bind-n-seq libraries were analyzed on an Illumina MiSeq-sequencing platform, aiming for an average of 30,000 reads per sample. For isolation of the pure insert sequence, the full MssI-restriction site (GTTT|AAAC) together with adjacent 3′-adapter sequence were trimmed from reads. In case of second selection round samples, the in vitro transcription derived 5′-GGG-triplet was included for trimming. Resulting sequences were filtered for length of the permutated RNA sequence library pools applied for target motif selection (8- or 14-mer). Remaining sequences were converted into fast format. Fasta files were analyzed with the original (for 14-mers) (Zambelli et al, 2014) or a suitably modified version (for 8-mers) of the Weeder2 software, including the identification of recurrent motifs of 5-, 6-, and 7-nt in length. For Weeder2 analysis, randomized input RNA pools were cloned and analyzed in parallel with endo-bind-n-seq libraries and served as an unbiased reference for background subtraction during motif enrichment calculations. Processing of sequencing output was performed with tools running inside our local galaxy instance. Cutadapt (v1.6) was used for adapter trimming and size filtering steps. Motifs were analyzed with Weeder2 and visualized using WebLogo3 server.

### Other methods

**Western blot analysis** To detect and analyze HA-tagged recombinant protein or immunoprecipitation reactions of FLAG-HA-tagged and endogenous proteins, as well as input lysates, samples were separated on a 10% (wt/vol) SDS-PAGE-gel. Proteins were subsequently transferred onto a nitrocellulose membrane (GE healthcare) by semi-dry blotting (20 V, 1.5 h). The membrane was blocked with 5% (wt/vol) milk powder in TBS-T at RT for 1 h and incubated with appropriate primary antibody overnight at 4°C with agitation. The membrane was washed three times with TBS-T before the cognate secondary antibody (1: 10,000 in 5% [wt/vol] milk powder in TBS-T) was applied for 1 h at RT. After washing again three times with TBS-T, fluorescent labels were detected using the Odyssey Infrared Imaging System (Li-COR). Applied antibodies are listed below.

Primary antibody.

| Primary antibody | | Dilution | Source |
| --- | --- | --- | --- |
| Anti-HA | HA.11, Clone 16B12; Covance | 1:1,000 | Mouse, mAb |
| Anti-HNRNPA1 | sc-32301, Clone 4B10; Santa Cruz | 1:1,000 | Mouse, mAb |
| Anti-CELF1 | 3B1, ab9549; Abcam | 1:1,000 | Mouse, mAb |
| Anti-GRSF1 | ERP16678, ab205531; Abcam | 1:1,000 | Rabbit, mAb |

Secondary antibody.

| Secondary antibody |
| --- |
| Goat polyclonal anti-Rabbit IgG, IRDye 800CW conjugated antibody (LI-COR) |
| Goat polyclonal anti-Mouse IgG, IRDye ye 800CW conjugated antibody (LI-COR) |

**Life Science Alliance**

## Quantification and statistical analysis

Western blot signals were detected and quantified with associated Application Software Version 3.0.30 of the Odyssey Infrared Imaging System 9120 (LI-COR Biosciences). Phosphor image signals were detected and quantified using the Quantity One analysis software Version 4.6.9 (Bio-Rad laboratories) with local background correction.

Signal intensity and protein concentration calculations, as well as standard deviations were calculated with Excel and associated Solver add-in program. A Chi-square Goodness-of-Fit Test was performed for approximation calculations to generate the absolute protein amount quantification-reference curve in Fig 5B. Therefore, Western blot signal intensities were quantified and defined as fixed y-axis factor. According to a one site dependent binding saturation, in which unspecific background binding is considered ($Y = B_{max} \times X / (X + K_d) + NS \times X + BG$; equation as provided by GraphPad Prism V5.04), the x-axis factor equaled our known absolute protein amount loaded in quantified Western blots, in which defined amounts of purified GST-HA-hnRNPA1(2–187) were titrated. By Chi-Square Test approximation, the additional equation factors were determined and the thereby completed equation was used to calculate back from quantified Western blot signals to absolute protein amounts in other experiments. As an internal reference, 1 $\mu$g of the reference protein GST-HA-hnRNPA1(2–187) were co-loaded at least in duplicates in independent Western blots for normalization of the quantified signals for subsequent protein mass calculations. Final protein concentrations in endo-bind-n-seq were determined in RNA binding reactions with the volume of 400 $\mu$l and the molecular weight of applied proteins.

## In vitro transcription of RNAs from primer templates

Specific RNA motifs flanked by random linker sequences were transcribed from DNA primer oligos with T7 RNA polymerase. Templates were prepared in a 5-cycle Phusion PCR (Thermo Fisher Scientific), applying 2 $\mu$M of universal T7 forward primer and 2 $\mu$M of RNA specific primer harboring the target motif (Metabion, Table S1). PCR products were directly used in a 500 $\mu$l transcription reaction (30 mM Tris pH 8.0, 10 mM DTT, 0.01% [vol/vol] Triton X-100, 25 mM MgCl$_2$, 2 mM spermidine, 30% [vol/vol] DMSO, 5 mM ATP, 5 mM CTP, 5 mM UTP, 5 mM GTP, 0.4 U/ml TIPP [NEB] and 0.1 mg/ml T7-RNAP) and incubated overnight at 37°C. Transcription reactions were separated on a 12% urea PAGE gel and visualized by UV shadowing. RNA products were cut out, eluted with water from crushed gel pieces under rotation overnight at 4°C and finally precipitated from supernatant by adding f.c. 500 mM NaCl and 0.7 volumes of isopropanol. Solutions were mixed, incubated at −20°C for at least 1 h and RNA was pelleted at 20,000 g at 4°C for 30 min. Pellets were washed with 80% (vol/vol) ethanol, dried at 60°C for 5–10 min, and dissolved in RNase-free water.

## Radioactive ZC3H7B RNA target competitor assay

1 mg of FH-ZC3H7B overexpression total lysate from HEK293T cells were incubated with 25 $\mu$l FLAG-M2 Agarose beads (Sigma-Aldrich) in a total volume of 1 ml, filled up with IP lysis buffer (50 mM Tris–HCl, pH 7.5, 300 mM NaCl, 1 mM AEBS, 1 mM DTT, 0.5% [vol/vol] NP-40), for 2 h at 4°C under rotation. Beads were washed twice with IP wash buffer 3 (500 mM NaCl, 50 mM Tris–HCl pH 7.5, 0.05% [vol/vol] NP-40, 1 mM DTT) and once with RNA binding buffer (150 mM KCl, 25 mM Tris–HCl pH 7.5, 3 mM MgCl$_2$, 0.01% [vol/vol] NP-40, 1 mg/ml BSA, 1 mM DTT, 5% [vol/vol] glycerol), and finally resuspended in 440 $\mu$l RNA binding buffer, aiming for a final protein concentration of 50–60 nM. 10% of the bead suspension was taken for Western blot analysis. Samples were supplemented with 1 pmol of $^{32}$P-labeled ZC3H7B target RNA (Motif 1 [AC]$_5$AGUUUCG [AC]$_5$, Motif 2 [AC]$_5$AUAGAU[AC]$_5$), combined either without or with 1, 2, or 4 pmol of unlabeled competitor RNA (Motif 1, Motif 2, or an unrelated control [AC]$_5$AGAGAG[AC]$_5$) and incubated for 30 min at RT agitating. Binding reactions were separated on a 12% urea PAGE and bound, labeled ZC3H7B target RNAs were analyzed by phosphor imaging.

## Radioactive filter binding assay

10,000 cpm of $^{32}$P-labeled ZC3H7B target RNA motif, a motif mutant and an independent ZC3H10 motif were diluted in 30 $\mu$l binding buffer (50 mM Tris–HCl pH 8.0, 150 mM NaCl, 5% glycerol) and incubated with different concentrations of purified ZC3H7B zinc-finger domain (aa 415–956) ranging from 0 to 800 ng. After incubation for 5 min at 20°C, the samples were filtered through a nitrocellulose membrane (0.45 mM; GE Healthcare) that was pre-equilibrated with binding buffer. The membrane was washed once with 20 ml of binding buffer and dried briefly. The spots corresponding to filtered samples were excised and analyzed by scintillation counting.

## Cloning and mammalian overexpression of proteins

For mammalian overexpression of FLAG-HA-tagged protein (from VP5 vector) and recombinant expression and purification of His$_6$-tagged (from pET32a) or GST-tagged (from pGEX-4T-1) protein in bacteria, vectors were cloned as listed in supplementary data (Table S1).

For overexpression of proteins, HEK293T cells were freshly plated on a 15 cm cell culture dish with 20% confluency. After 2–4 h, 10 $\mu$g of cloned VP5 plasmid DNA were transfected, using the calcium phosphate precipitation method. Cells were harvested after 48 h and directly used for IP experiments or stored as pellets at −80°C.

## Recombinant protein purification

Recombinant proteins were purified upon transformation and expression in *E.coli* BL21(DE3) cultures overnight at 25°C. GST-ZC3H7B expression was induced by autoinduction according to (Studier, 2005), using combined 5052x and M-supplement. Expression of all other protein constructs (GST-tagged CELF1, GRSF1, HA-hnRNPA1, hnRNPA1, ZC3H10, and His-tagged hnRNPA1) was canonically induced by addition of 1 mM IPTG at an OD$_{600}$ of 0.6. For purification of GST-fusion protein, bacteria were lysed by sonication in GST-lysis buffer (PBS supplemented with 1 M NaCl and 2 mM DTT), and lysate was cleared by centrifugation (50,000*g*, 30 min, 4°C). Supernatant was passed through a 0.45 mM filter membrane (Roth) before loading on

a 5 ml *GSTrap column* (GE Healthcare). After extensive washing with lysis buffer, bound GST-fusion protein was eluted with 10 mM Glutathione in PBS supplemented with 50 mM Tris pH 8. The eluate was concentrated to a volume of 0.5–1 ml with a *Vivaspin 20* ultrafiltration device (MWCO 10,000 or 30,000, Sartorius) and loaded on a *Superdex 200 10/30 GL* column (GE Healthcare) equilibrated with 50 mM HEPES pH 7.5, 200 mM NaCl and 1 mM DTT. Peak fractions were pooled, concentrated, and adjusted to 40% (vol/vol) glycerol for storage at −80°C. For purification of untagged protein, procedure was analogous as described above, however including removal of the GST-tag by cleavage at an additional TEV-site of the downstream protein coding sequence. For this, after the first *GSTrap column* run, 0.1 mg/ml of GST-tagged TEV protease was added to the pooled fractions, incubated overnight at 4°C and finally buffer was exchanged back to GST-lysis buffer during the concentration step. Samples were subsequently reloaded on a 5 ml *GSTrap column*, this time collecting the flow through fractions containing the untagged protein. Fractions were concentrated and further processed as described above. 6x His-tagged hnRNPA1 was purified as described for GST-fusion proteins, however using 50 mM Na-phosphate pH 8, supplemented with 1 M NaCl and 10 mM Imidazole as lysis buffer. Cleared and filtered lysates were loaded on an equilibrated, Ni-charged IMAC-Sepharose column (GE Healthcare) and the protein was eluted with 50 mM Na-phosphate pH 8, 300 mM NaCl and 500 mM Imidazole. All further steps were performed as described above.

### Heterologous overexpression of SmAP1 in *E. coli* and affinity purification

Heterologous overexpression of *P. furiosus* SmAP1 fused with a StrepII-tag sequence to the C-terminus in *E. coli* JW4130Δhfq (Keio Collection) and affinity purification were carried out similar as described previously (Reichelt et al, 2023). Briefly, 1 liter LB medium containing 25 μg/ml kanamycin and 100 μg/ml ampicillin were inoculated to reach a final OD600 of 0.1–0.2. Cultures were grown at 37°C to an OD of 0.5–0.6. The culture was cooled to 18°C and expression of SmAP1 was induced by adding 0.5 mM IPTG. After growth overnight at 18°C, the cells were harvested and the cell pellet stored at −20°C. A cell pellet corresponding to 350 ml overexpression culture was resuspended in 1 ml Strep100 buffer (10 mM Tris–HCl pH 8.0, 100 mM NaCl, 10% [vol/vol] glycerol), lysozyme was added and incubated for 30–60 min on ice. The cells were lysed for 5 × 3 min by sonication using a Bandelin Electronic Sonopuls HD 2070 homogenizer (cycle: 80%, power 50%). The lysate was treated with 2 μl Benzonase Nuclease (>250 U/μl) (Merck) and 25 μl RNase Cocktail enzyme mix (Invitrogen) at 37°C for 1 h, 64°C for 20 min and afterward centrifuged at 20,000$g$ for 30 min at 4°C. After centrifugation, the supernatant was immediately transferred in fresh tubes, filtered (0.2 μm) and stored at 4°C. 1 ml of the supernatant was incubated with 100 μl MagStrep "type3" XT beads (IBA Lifesciences) for 30 min at 4°C. Beads were immobilized using a magnetic rack and consecutively washed (0.2 ml) 5× with Strep100 buffer followed by five washing steps with Strep2000 buffer (10 mM Tris–HCl pH 8.0, 2000 mM NaCl, 10% [vol/vol] glycerol) and again, five washing steps with Strep100 buffer. SmAP1 was eluted with 50 μl 1x BXT elution buffer (IBA Lifesciences) at 4°C for 10 min and stored at 4°C.

### Electro mobility shift assays

Assays were carried out using 10 nM 5′ Hexachloro-fluorescein (HEX) – labeled 19 nt RNA fragments (SmAP1 motif: 5′ GAAUUU-GAGUUUAAUGAAC -3′; C-rich control: 5′ GAACCCCCCCCCCAAUGAAC -3′) increasing amounts of SmAP1 heptamer (nM: 0, 1.25, 2.5, 5, 10, 20, 25, 50, 100, 200, 400) in binding buffer (20 mM HEPES–KOH pH 7.4, 100 mM NaCl, 1 mM MgCl$_2$, 0.1% [vol/vol] Tween20) supplemented with 40 mM EDTA and 10% (vol/vol) glycerol and incubated at 70°C for 10 min. Samples were separated by electrophoresis (150 mV, 40 min) using a Tris/Boric Acid/EDTA (TBE) buffer system and a native 6% TBE gel.

## Data Availability

Sequencing data are available at the SRA database under the accession: PRJNA1097567, Temporary Submission ID: SUB14331669.

## Supplementary Information

## Acknowledgements

We thank S Ammon and C Friederich for technical support. We thank J Heimbucher for providing mouse tissues. This work was supported by the Deutsche Forschungsgemeinschaft (Priority program SPP 1935 "Dciphering the mRNP code," Me 2064/6-1/2; DIP project FI 573/26-1). G Meister was supported by the BMBF in the framework of the Cluster4Future program (Cluster for Nucleic Acid Therapeutics Munich, CNATM) (Project ID: 03ZU1201BD).

## Author Contributions

TN Hanelt: resources, formal analysis, validation, investigation, and writing—original draft.
N Treiber: conceptualization, formal analysis, and investigation.
T Treiber: conceptualization, formal analysis, and investigation.
G Lehmann: data curation, software, and formal analysis.
N Eichner: resources, data curation, and software.
T Rothmeier: formal analysis and investigation.
G Schmid: formal analysis and investigation.
R Reichelt: conceptualization, data curation, formal analysis, and investigation.
Zambelli F: resources, data curation, and software.
G Pavesi: resources, data curation, software, and supervision.
D Grohmann: conceptualization, supervision, funding acquisition, validation, and project administration.
G Meister: conceptualization, supervision, funding acquisition, validation, visualization, and writing—original draft, review, and editing.

**Conflict of Interest Statement**

The authors declare that they have no conflict of interest.

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
