## [Reviewer comments · Life Science Alliance]

Life Science Alliance

Endo-bind-n-seq: identifying RNA motifs of RNA binding proteins isolated from endogenous sources

Gunter Meister, Tiana Hanelt, Nora Treiber, Thomas Treiber, Gerhard Lehmann, Norbert Eichner, Tamara Rothmeier, Georg Schmid, Robert Reichelt, federico zambelli, Giulio Pavesi, and Dina Grohmann

DOI: <https://doi.org/10.26508/lsa.202402782>

Corresponding author(s): Gunter Meister, University of Regensburg

Review Timeline:

Submission Date:	2024-04-19
Editorial Decision:	2024-06-10
Revision Received:	2024-11-05
Editorial Decision:	2024-11-06
Revision Received:	2024-11-14
Accepted:	2024-11-15

Transaction Report:

June 10, 2024

Re: Life Science Alliance manuscript #LSA-2024-02782

Prof. Gunter Meister
University of Regensburg
Regensburg 93053
Germany

Dear Dr. Meister,

Thank you for submitting your manuscript entitled "Endo-bind-n-seq: identifying RNA motifs of RNA binding proteins isolated from endogenous sources" to Life Science Alliance. The manuscript was assessed by expert reviewers, whose comments are appended to this letter. We invite you to submit a revised manuscript addressing the Reviewer comments.

Thank you for this interesting contribution to Life Science Alliance. We are looking forward to receiving your revised manuscript.

Sincerely,

B. MANUSCRIPT ORGANIZATION AND FORMATTING:

Reviewer #1 (Comments to the Authors (Required)):

The authors report on improving RNA Bind-n-Seq (Lambert et al. 2014), a method to identify RNA motif interacting with RNA binding proteins (RBPs) or domains thereof. Essentially, the method selects for interaction of recombinant or native RBPs with a 8-mer and/or 14 nts long RNAs in a pool after affinity selection and sequencing. This "improved" method includes a second selection step that shall improve selectivity; and they established its applicability with native RBPs captured from cell lysates, and hence calling the method 'endo-bind-n-seq'. The top motifs were identified bioinformatically with Weder from sequencing data of the selected RNA pool. More specifically, the authors modified the first selection step of the original RNA Bind-n-Seq by adding a 5' RNA adaptor, which contains T7 RNA promoter region at its 3' end for transcription, and a 3' DNA adaptor to introduce a restriction site, into their 8/14 nt selected RNA pools after selection with the RBP. The ligated product was reverse transcribed, PCR amplified and gel purified and sequenced. The adaptors were then cleaved off with a restriction enzyme (MssI), and cleaved product in vitro transcribed to generate a second RNA pool for affinity selection and sequencing.

To evaluate the system, the authors tested two recombinant RBPs, hnRNPA1 and ZC3H10, to evaluate the method in the first instance. They found similar motif sequences in RNA Bind-n-Seq compared to SELEX and RNA compete methods once they used their 8-nt RNA library. The advantage of 2 rounds of selection was tested with a non-canonical RBPs ZC3H7B, and SmAP1, showing higher scores for specific motifs after two rounds and results were confirmed with filter-binding and EMSA experiments. They further tested applicability of the method to affinity purified or IPed RBPs from cell extracts as compared to recombinant proteins, showing some differences between recombinant proteins compared to cell expressed proteins for some RBPs (e.g. hnRNPA1), while others gave consistent results (GRSF1). Furthermore, it was noted that during IPs associated proteins can co-IP RNA targets and so, the selected RNA motif may reference to a co-IPed protein (CSTF1 co-IPs CSTF2). Finally, they showed the dependence of interactions on protein concentration on beads of in extract and the dependence on salt concentration, indicating that the approach needs optimisation for each individual RBP under consideration.

Overall, this is an interesting body of work with detailed methods description which could be of interest to other researchers in the field. While it certainly provides an alternative to previously established methods, we had the impression that the advantages of this approach seem not as compelling as the authors claim. For instance, the author mention limitations of other methods (SELEX and CLIP) relate to extensive sequencing or bioinformatic data processing. However, 'endo-bind-n-seq' method also needs substantial optimisations (concentration, buffers), includes several steps of RNA/DNA modifications (ligation, digests, etc), and it requires extensive sequencing and data processing. Hence, the claimed advantages to previous approaches are not immediately obvious (abstract, introduction lines 113 to118) and the authors should revise respective statements accordingly.

Specific comments:

1-The introduction is rather lengthy and could be shortened.

2-Line 136. The authors claim "homogenous base distribution at all position (Suppl. Fig. 1A)". It though looks like that this is not the case as C's are underrepresented from base 2-7 and likewise for other bases at other positions. The authors should modify this sentence accordingly. Furthermore, the authors claim to have optimised the "synthesis conditions"? It is unclear how this was done and how it improved the RNA pool. This should be explained in main or added to a Supplement.

3-Line 143. The reverse transcribed cDNA product is PCR amplified and run on an acrylamide gel. The authors should show a picture of that gel in the supplementary figures so the reader can judge the efficiency of their modified first step methodology.

4-Fig. 1B,C. The authors tested the performance with recombinant proteins after 2 rounds of selection. This seem to work well but establish the advantage for 2 rounds of selection, the authors should also show the output from the first round of selection (Weder). Furthermore, the authors only displayed the top hit from Weder in the figures. Here, the authors should add all of top motifs with a given consistent threshold in the supplementary materials/ or even provide the original output.

5-Limitations using the 14 nts library should be more clearly stated. Specificity drop Fig. 1C should be discussed.

6-Fig. 2D. Control gel (right side) seems to be less exposed than the SmAP1 motif reaction. If possible, a longer expounded image should be added.

7-Line 205. The authors should explain rationale for selection those RBPs for their tests.

8-Line 214. "For CELF1 and GRSF1, retrieved motifs were identical...between the used methods". This statement seems not valid for CELF1. The authors should modify the statement accordingly i.e. retrieved similar motifs and place the Weder output with motifs in the Supplement (see comment above).

9-Lines 224-227/ Fig. 3F. The authors may have isolated two different hnRNPA1 isoforms from brain and liver. Intriguingly, HRNAPA1 sequences selected with IP from cell lines cells gave different results (Fig. 3D); while the output from tissues resembles the motif selected with the recombinant protein. The possibility for different isoforms (difference in Western) should be discussed; and the authors should add a likewise Western for IPs displayed in Fig. 3D.

10-Line 237. The 14 nt RNA library was not as successful in motif selection (lower significance), so using that library for complex RNA binding affinity may not be advisable. The differences between the two libraries should be better discussed.

11-It would be beneficial if authors could check and update references with more recent reviews/research articles on the specific proteins under study.

Reviewer #2 (Comments to the Authors (Required)):

The manuscript by Hanelt et al. modified the RNA bind-n-seq method by enabling two rounds of in vitro RNA selection and compared the enriched RNA motifs using recombinant RBPs purified from *E. coli* with those using RBPs immunoprecipitated from transfected cells or endogenous RBPs where they named it as Endo-bind-n-seq. The authors showed that two rounds of RNA selection yielded clear, strong enrichment of RNA motifs and the use of RBP immunoprecipitated from cell extracts yielded similar RNA motifs as recombinant RBP, indicating that RBP IPed from cell extracts can be an alternative source when recombinant RBP is not available. Furthermore, the authors showed that IPed RBP can identify indirect RNA motifs from its binding partners and likely from a larger complex containing several RBPs although the latter is not further explored in the study. Most results supported the conclusion. However, many negative controls were not presented in the figures. This manuscript is methodology study that is helpful for audience in the field of RNA-RBP interaction and is very useful to identify RNA motifs for a particular RBP.

Major points:

1. Enriched motifs from proper controls, e.g., GST and Ni-LMAC beads only, should be presented in Figures 1B and 1C. Comparison of enriched motifs from 1 and 2 rounds of RNA selection should be included in Figures 1B and 1C to demonstrate that 2-round selection is better than 1 round.
2. Negative control should be included in Figure 2C. Why 10-mer RNA oligo was using in Figure 2B as the authors mentioned "the most enriched 8-mer sequence", line 185? A competition using cold RNA should be shown in Figure 2D to show that the binding is specific.
3. Motifs enriched using proper negative controls should be included in Figures 3B, 3C, 3D and 3E. Control IP using IgG is needed in Figure 3F. The binding of CELF1 and GRSF to enriched motifs needs to be verified by another RNA binding assay.
4. A proper control should be included in Figure 4A. It will be better to identify enriched motifs in the cells expressing FH-CSTF1 without the expression of CSTF2, e.g., siRNA knockdown, to show that the binding presented in Figure 4B is via CSTF2. The binding of FH-Pumilio 1 and FH-CSTF1 to identified motifs needs to be verified by another RNA binding assay.
5. In Figure 4C, enriched motifs using FH-ZC3H7B should be compared to those from full-length GST- ZC3H7B. Alternatively, GST-ZC3H7B(415-956) compares to FH- ZC3H7B(415-956). The difference between Figure 2C and Figure 4C may be because of the protein sources, *E. coli* vs. mammalian cells.
6. The authors' conclusion that "Furthermore, while both unlabeled RNAs competed for binding, the motif identified with the recombinant Zn fingers competed better, which was rather unexpected.", line 263-264, is too premature. 32P-AUAGAU vs unlabeled AUAGAU and 32P-AGUUUCG vs. unlabeled AGUUUCG should be performed. In addition, higher folds (30-100) of competitors should be included.
7. Control IP should be included in Figure 6B.

Minor points:

1. The lengthy description regarding RBPs in the first paragraph of Introduction is not very relevant to the scope and purpose of the study.
2. The authors refer to RNA sequences bound by RBPs as "RNA-binding motifs" is very confusing. Normally, RNA-binding motifs is referred to the domains in RBPs that bind to RNA. "RNA motifs" recognized by RBP is a better term.
3. What is "increased intensity output" in Figure 5A?
4. In Figure 5C for FH-ZC3H7B quantification, what is the input in lanes 9 and 10, GST-HA-hnRNPA1 or GST-HA-ZC3H7B?

Response to the reviewers

We would like to thank the reviewers for their very constructive criticism that helped us to improve our manuscript. We hope that we have satisfied the reviewers' concerns with the revised version of our manuscript.

We respond to the reviewers' comments as follows:

Reviewer #1*Main comment:*

Overall, this is an interesting body of work with detailed methods description which could be of interest to other researchers in the field. While it certainly provides an alternative to previously established methods, we had the impression that the advantages of this approach seem not as compelling as the authors claim. For instance, the author mention limitations of other methods (SELEX and CLIP) relate to extensive sequencing or bioinformatic data processing. However, 'endo-bind-n-seq' method also needs substantial optimisations (concentration, buffers), includes several steps of RNA/DNA modifications (ligation, digests, etc), and it requires extensive sequencing and data processing. Hence, the claimed

advantages to previous approaches are not immediately obvious (abstract, introduction lines 113 to 118) and the authors should revise respective statements accordingly.

We thank the reviewer for the positive evaluation of our manuscript. We would rather see endo-bind-n-seq as a highly valuable complementation to other methods, which all have their own limitations or difficulties (including our endo-bind-n-seq). We think one of the main advantages of our approach is that endogenous proteins can be used for RNA binding motif selection instead of recombinant proteins, which might not be easily available in all labs and for all RBPs. Compared to CLIP experiments, data analysis and validation of our method is much simpler. Of course, CLIP experiments can answer additional questions, when carefully carried out (target RNA identity, for example). Furthermore, we included a thorough evaluation of experimental conditions to also raise awareness that RBPs have individual biophysical properties, which needs to be considered. Nevertheless, we agree with the reviewer that we may have overstated our method particularly in the abstract and toned down our statements.

Specific comments:

1-The introduction is rather lengthy and could be shortened.

Since our method aims at an audience not deeply involved in RBPs and RNA biology (e.g. developmental biologists, who study specific processes), we would like to keep the rather detailed description of the different methods that are available (including potential challenges). We agree that the first RBP part is somewhat lengthy and shortened it as suggested.

2-Line 136. The authors claim "homogenous base distribution at all position (Suppl. Fig. 1A)". It though looks like that this is not the case as C's are underrepresented from base 2-7 and likewise for other bases at other positions. The authors should modify this sentence

accordingly. Furthermore, the authors claim to have optimised the "synthesis conditions"? It is unclear how this was done and how it improved the RNA pool. This should be explained in main or added to a Supplement.

Thank you very much for pointing out these two imprecise statements. The RNA library was generated by a company inserting nucleotides randomly at each position. After we received the library, we sequenced it and found that some nucleotides were under- or overrepresented at distinct positions. Thus, the company re-synthesized it and adjusted the concentrations of the nucleotides for each position accordingly until we reached a library with a nearly homogenous nucleotide distribution. We have added this information now to the Methods section of our manuscript. We agree that our statement of homogenous distribution is not fully accurate and thus we have revised it. We now state: "We validated the base distribution in the input pool by RNA-seq and optimized the synthesis conditions to reach a homogenous base distribution at all positions, which was unfortunately not fully achieved towards the 3' end of the 8-mer library, while the nucleotides of the 14-mer library are almost equally distributed (Suppl. Fig. 1A)."

3-Line 143. The reverse transcribed cDNA product is PCR amplified and run on an acrylamide gel. The authors should show a picture of that gel in the supplementary figures so the reader can judge the efficiency of their modified first step methodology.

Since we are presenting a new methodological approach, which we hope that will be used in many labs in the future, we agree that a more detailed description is indeed valuable. Therefore, we have added several images of steps during the workflow of endo-bind-n-seq to the supplementary material as suggested by the reviewer (new suppl. Figure 2).

4-Fig. 1B,C. The authors tested the performance with recombinant proteins after 2 rounds of selection. This seem to work well but establish the advantage for 2 rounds of selection, the

authors should also show the output from the first round of selection (Weeder). Furthermore, the authors only displayed the top hit from Weeder in the figures. Here, the authors should add all of top motifs with a given consistent threshold in the supplementary materials/ or even provide the original output.

We agree with reviewer#1 that a broader presentation of Weeder output sequences allows for a better view on our data analysis platform and will be beneficial for users. Therefore, we have included the top-scoring sequences of experiments shown in Figure 1 to the supplements of our manuscript (suppl. Fig. 3). Of note, all primary sequencing data are publicly available as indicated in the “data availability” section.

5-Limitations using the 14 nts library should be more clearly stated. Specificity drop Fig. 1C should be discussed.

We hypothesize that a 14 nt library could either fold and generate secondary structures stronger than an 8-mer library. Such structures might be sequence specific and thus some sequences may not be available for our selection reaction. A second hypothesis would be that RBPs often have rather short binding motifs (4-5 nts). If such motifs overlap multiple times on one oligomer, it may be difficult to retrieve from the sequencing data and the discovered motif becomes more imprecise and scores drop. We made this clearer in the revised text. We state now: “This is most likely due to the presence of multiple overlapping binding sites that are present on one read. Sequences in such reads may be difficult to define and therefore scores drop. In addition, 14-mers might form more secondary structures, which could also affect data analysis. “

6-Fig. 2D. Control gel (right side) seems to be less exposed than the SmAPI motif reaction. If possible, a longer exposed image should be added.

As suggested by reviewer#1, we have now included a longer exposure of the control gel in

order to make it better comparable with the gel on the left side. Thanks for pointing this out.

7-Line 205. The authors should explain rationale for selection those RBPs for their tests.

Since our study primarily focuses on the improvement of bind-n-seq and to expand it to beads-bound proteins from endogenous sources, we used for most of our work well-characterized RBPs and assessed whether we can retrieve the well-known binding motifs. This allowed us to focus on the method rather than the identification of novel binding motifs. We included “well-characterized” into the sentence in line 205.

8-Line 214. "For CELF1 and GRSF1, retrieved motives were identical...between the used methods". This statement seems not valid for CELF1. The authors should modify the statement accordingly i.e. retrieved similar motifs and place the Weeder output with motifs in the Supplement (see comment above).

Thank you for pointing this inaccurate statement out and suggesting a better wording. We have changed the text accordingly and also present the top Weeder scores as requested in our new supplementary Figure 4.

9-Lines 224-227/ Fig. 3F. The authors may have isolated two different hnRNPA1 isoforms from brain and liver. Intriguingly, HRNAPA1 sequences selected with IP from cell lines cells gave different results (Fig. 3D); while the output from tissues resembles the motif selected with the recombinant protein. The possibility for different isoforms (difference in Western) should be discussed; and the authors should add a likewise Western for IPs displayed in Fig. 3D.

We agree with the reviewer that different isoforms could be captured from different tissues and we mention that in our discussion. However, to make such a point, we would need to go much deeper and validate this much better. Therefore, we have removed the western blot (we

do not even know whether these are isoforms or modifications etc.). We feel it is still too preliminary to be shown as part of this manuscript.

10-Line 237. The 14 nt RNA library was not as successful in motif selection (lower significance), so using that library for complex RNA binding affinity may not be advisable. The differences between the two libraries should be better discussed.

We would like to refer to our response to point 5 above. We have now explained the rationale behind this approach better.

11-It would be beneficial if authors could check and update references with more recent reviews/research articles on the specific proteins under study.

When introducing the RNA motifs of the proteins under study, we attempt to cite the first study that identified the binding motif, which we consider to be fair to the authors. More recent work often puts these proteins into specific pathologies and cellular functions without refining the binding motif etc. Therefore, we would like to remain with this strategy. When introducing general RBP families and properties, we cite recent reviews from 2023 for example.

Reviewer #2

The manuscript by Hanelt et al. modified the RNA bind-n-seq method by enabling two rounds of in vitro RNA selection and compared the enriched RNA motifs using recombinant RBPs purified from E. coli with those using RBPs immunoprecipitated from transfected cells or endogenous RBPs where they named it as Endo-bind-n-seq. The authors showed that two rounds of RNA selection yielded clear, strong enrichment of RNA motifs and the use of RBP immunoprecipitated from cell extracts yielded similar RNA motifs as recombinant RBP, indicating that RBP IPed from cell extracts can be an alternative source when recombinant

RBP is not available. Furthermore, the authors showed that IPed RBP can identify indirect RNA motifs from its binding partners and likely from a larger complex containing several RBPs although the latter is not further explored in the study. Most results supported the conclusion. However, many negative controls were not presented in the figures. This manuscript is methodology study that is helpful for audience in the field of RNA-RBP interaction and is very useful to identify RNA motifs for a particular RBP

We thank the reviewer for the positive evaluation of our manuscript.

Major comments:

1. Enriched motifs from proper controls, e.g., GST and Ni-LMAC beads only, should be presented in Figures 1B and 1C. Comparison of enriched motifs from 1 and 2 rounds of RNA selection should be included in Figures 1B and 1C to demonstrate that 2-round selection is better than 1 round.

As also requested by reviewer#1, we have now added the top-scoring Weeder output sequences of several experiments, which would allow for a broader view on our results. This will be beneficial for future users of endo-bind-n-seq. We have included these data as new supplementary Figure 3.

2. Negative control should be included in Figure 2C. Why 10-mer RNA oligo was using in Figure 2B as the authors mentioned "the most enriched 8-mer sequence", line 185? A competition using cold RNA should be shown in Figure 2D to show that the binding is specific.

Our phrasing is indeed unclear and we apologize that we failed to explain this properly. First, 10-mer RNAs were used for this specific filter binding assay, which is different to the two libraries (8-mer, 14-mer) that we used in endo-bind-n-seq. Second, with "8-mer" we referred to the motif that we initially obtained from endo-bind-n-seq, which was somewhat mis-

leading. We have rephrased this now and state as follows: “Recombinant ZC3H7B(415-956) shows a strong binding preference to an RNA fragment containing the identified AUAG sequence (AUAGUGUAGU) compared to a sequence with a mutant motif (AAAGUGAAGU) and an unrelated control (Fig. 2B) corroborating our endo-bind-n-seq results.”

Since the binding of SmAP1 to the RNA motif was so specific, with no binding observed to a control sequence, we did not include a cold competitor RNA. We agree that such an experiment would be very helpful when kinetics and affinities will be studied in future work. However, we feel that such elaborate experiments are beyond the scope of this manuscript.

3. Motifs enriched using proper negative controls should be included in Figures 3B, 3C, 3D and 3E. Control IP using IgG is needed in Figure 3F. The binding of CELF1 and GRSF to enriched motifs needs to be verified by another RNA binding assay.

We agree that the mentioned negative control would be very helpful if the main aim of our work was to functionally and in-depth characterize the interaction of CELF1 and GRSF1 with their putative natural RNA targets. However, we intended to select already well-known and well-characterized RBPs, which would allow us to focus on the development and further improvement of bind-n-seq. We would like to mention that in our view a main advantage of our method is the use of isolated proteins from endogenous sources. We thus focused on the known and in the literature well-characterized target motifs and optimized our method to retrieve these motifs from different sources.

4. A proper control should be included in Figure 4A. It will be better to identify enriched motifs in the cells expressing FH-CSTF1 without the expression of CSTF2, e.g., siRNA knockdown, to show that the binding presented in Figure 4B is via CSTF2. The binding of FH-Pumilio 1 and FH-CSTF1 to identified motifs needs to be verified by another RNA

binding assay.

Similarly to our response to point 3, the PUM binding motif is one of the most studied RNA binding motifs and thus we would like to refer to the literature rather than repeating such experiments. The reviewer suggests an interesting experiment, which would clearly further validate our findings. However, structural data confirmed that the two proteins are in a tight complex and CSTF2 binds the retrieved motif in these complexes. Based on these data, we think it is fair to conclude that binding is indeed mediated through CSTF2 in our endo-bind-n-seq experiments. Knock down of CSTF2 may cause additional effects (for example, potential destabilization of the complex, generating unspecific interactions of CSTF1 in the absence of its binding partner) and, although highly interesting, a deeper study of this complex is not in the focus of our method development.

5. In Figure 4C, enriched motifs using FH-ZC3H7B should be compared to those from full-length GST-ZC3H7B. Alternatively, GST-ZC3H7B(415-956) compares to FH-ZC3H7B(415-956). The difference between Figure 2C and Figure 4C may be because of the protein sources, E. coli vs. mammalian cells.

Thank you very much for pointing this out. Please see also our response to point 6 below. We initially thought that the different protein variants may cause the slightly different motifs. However, using the short version immunoprecipitated from lysates did not select the motif shown in Fig. 4C, which is still puzzling. We tried very hard for a very long time to disentangle these findings but did not succeed (see also below). Unfortunately, we failed to produce recombinant full-length protein in bacteria and insect cells making interpretations even more difficult. Nevertheless, we would like to keep it as an example that particularly for binders with lower affinity, results are not always black or white and might be more complex. We discuss this now in more detail.

6. The authors' conclusion that "Furthermore, while both unlabeled RNAs competed for binding, the motif identified with the recombinant Zn fingers competed better, which was rather unexpected.", line 263-264, is too premature. 32P-AUAGAU vs unlabeled AUAGAU and 32P-AGUUUCG vs. unlabeled AGUUUCG should be performed. In addition, higher folds (30-100) of competitors should be included.

We understand that our results on the RNA motif of ZC3H7B appears premature. On the one hand, we identified an RNA motif with a recombinant fragment containing the Zn fingers and on the other hand we retrieved a slightly different motif with the full-length protein immunoprecipitated from cell lysates. To be honest, we tried very hard for a long time to mechanistically understand these findings but could not come to a satisfying conclusion. Since we present a novel method, we reasoned that showing such results may help future users and guide through experiments where proteins may have lower affinities to RNAs in a natural environment or maybe even cell stage-specific functions. Particularly when lower affinities are studied, results are never black or white and thus showing this and being maximal transparent would be beneficial for the field. It may really help to estimate results and prevent misinterpretations. We therefore hope that reviewer#2 accepts that we would leave the data to show that results are not always clear and easy to interpret.

We changed the text accordingly and now state: "Furthermore, the unlabeled RNA motif identified with the recombinant Zn fingers competed better, which was rather unexpected and still remains puzzling. These results may suggest that different motifs are selected under different experimental conditions or under different protein or RNA concentrations. Our data further underscore that not all RNA binding activities can easily be assessed using in vitro selection methods and particularly low affinity binders may generate motifs with low confidence that need to be thoroughly validated in independent in vivo experiments."

7. Control IP should be included in Figure 6B.

We used un-transfected HEK 293 cell lysates as control and these western blots do not show any signal.

Minor points:

1. The lengthy description regarding RBPs in the first paragraph of Introduction is not very relevant to the scope and purpose of the study.

We agree that our introduction was rather lengthy and we have shortened the description of RBPs as suggested.

2. The authors refer to RNA sequences bound by RBPs as "RNA-binding motifs" is very confusing. Normally, RNA-binding motifs is referred to the domains in RBPs that bind to RNA. "RNA motifs" recognized by RBP is a better term.

Thank you very much for carefully evaluating our manuscript. Indeed, RNA-binding motifs might be somewhat misleading. While a very common RNA binding domain in proteins is "RNA recognition motif" RRM, it is highly similar and thus we agree that RNA motif is the clearer term. We have changed the wording throughout the text as suggested by the reviewer.

3. What is "increased intensity output" in Figure 5A?

With "increased intensity output" we referred to the output of our imaging system, which might appear somewhat random. We have now changed the description to "increased signal intensity", which might be better accessible for readers. Thanks for pointing this out.

4. In Figure 5C for FH-ZC3H7B quantification, what is the input in lanes 9 and 10, GST-HA-hnRNPA1 or GST-HA-ZC3H7B

Both lanes and upper and lower panels show 1 μ g GST-HA-hnRNPA1(2-187) since we used this defined protein amount for normalization. We re-arranged the Figure to make this clearer.

November 6, 2024

RE: Life Science Alliance Manuscript #LSA-2024-02782R

Prof. Gunter Meister
University of Regensburg
Regensburg 93053
Germany

Dear Dr. Meister,

Thank you for submitting your revised manuscript entitled "Endo-bind-n-seq: identifying RNA motifs of RNA binding proteins isolated from endogenous sources". We would be happy to publish your paper in Life Science Alliance pending final revisions necessary to meet our formatting guidelines.

- please be sure that the authorship listing and order is correct
- please add the Twitter handle of your host institute/organization as well as your own or/and one of the authors in our system
- please add the author contributions to the main manuscript text
- please add a figure callout for your Figure S4 to the main manuscript text

A. FINAL FILES:

B. MANUSCRIPT ORGANIZATION AND FORMATTING:

Sincerely,

November 15, 2024

RE: Life Science Alliance Manuscript #LSA-2024-02782RR

Prof. Gunter Meister
University of Regensburg
Regensburg 93053
Germany

Dear Dr. Meister,

Thank you for submitting your Methods entitled "Endo-bind-n-seq: identifying RNA motifs of RNA binding proteins isolated from endogenous sources". It is a pleasure to let you know that your manuscript is now accepted for publication in Life Science Alliance. Congratulations on this interesting work.

DISTRIBUTION OF MATERIALS:

Again, congratulations on a very nice paper. I hope you found the review process to be constructive and are pleased with how the manuscript was handled editorially. We look forward to future exciting submissions from your lab.

Sincerely,
